

# Quasi 18-hour wave activity in ground-based observed mesospheric H$_2$O over Bern, Switzerland

Martin Lainer[1], Klemens Hocke[1,2], Rolf Rüfenacht[1,3], Franziska Schranz[1], and Niklaus Kämpfer[1,2]

[1]Institute of Applied Physics, University of Bern, Bern, Switzerland
[2]Oeschger Center for Climate Change Research, University of Bern, Bern, Switzerland
[3]Actual affiliation: Leibniz-Institute of Atmospheric Physics, Kühlungsborn, Germany

*Correspondence to:* M. Lainer (martin.lainer@iap.unibe.ch)

**Abstract.** Observations of oscillations in the abundance of middle atmospheric trace gases can provide insight into the dynamics of the middle atmosphere. Long term, high temporal resolution and continuous measurements of dynamical tracers within the strato- and mesosphere are rare, but would be important to better understand the impact of planetary and gravity waves on the middle atmosphere. Here we report on water vapor measurements from the ground-based microwave radiometer

MIAWARA located close to Bern during two winter periods of 6 months from October to March. Oscillations with periods between 6 and 30 hours are analyzed in the pressure range 0.01–10 hPa. Seven out of twelve months have the highest wave amplitudes between 15 and 21 hour periods in the mesosphere above 0.1 hPa. The quasi 18-hour wave is studied in more detail. We examine the temporal behavior and use SD-WACCM simulations for comparison and to derive characteristic wave features considering low-frequency gravity-waves being involved in the observed water vapor oscillations. The 18-hour wave

is also found in SD-WACCM horizontal wind data and in measured zonal wind from the microwave Doppler wind radiometer WIRA. For two cases in January 2016 we derive the propagation direction, intrinsic period, horizontal and vertical wavelength of the model resolved 18-hour wave. A south-westward to westward propagation with horizontal wavelengths of 1884 and 1385 km and intrinsic periods close to 14 h are found. Vertical wavelengths are below 6 km. We were not able to single out a distinct temporal correlation between 18-hour band-pass filtered water vapor and wind data time series, although H$_2$O should

mostly be dynamically controlled in the mesosphere and sub-diurnal time range. More sophisticated numerical model studies are needed to uncover the manifold effects of gravity waves on the abundance of chemical species.

## 1  Introduction

The dynamics of the middle atmosphere is controlled by a manifold spectrum of waves. Knowledge about the wave characteristics and incidence is important, not only to better understand the elements of middle atmospheric dynamics, but carrying

on to improve predictions of weather (Hardiman et al., 2011) and climate (Orr et al., 2010) models. Latter are getting more important since the social impact of severe weather events and climate change is increasing.

Waves with horizontal wavelengths reaching thousands of kilometers and showing periods up to several weeks are classified as planetary waves. A well-known class of planetary waves are Rossby waves (Salby, 1981b). Their periods range from 2 to approximately 18 days in the middle atmosphere, showing strong inter-annual variability (Jacobi et al., 1998). Investigations of




the quasi 2-day wave are found for instance in studies by Salby (1981a); Rodgers and Prata (1981); Yue et al. (2012) and more recently by Tschanz and Kämpfer (2015). Latter analyzed the 2-day wave signatures in arctic middle-atmospheric water vapor measurements in conjunction with the occurrence of sudden stratospheric warmings. Characteristics of the 5-day wave were analyzed by Rosenlof and Thomas (1990); Wu et al. (1994); Riggin et al. (2006); Belova et al. (2008) and waves with even longer periods have been observed in the mesosphere and lower thermosphere (Forbes et al., 1995; McDonald et al., 2011; Scheiben et al., 2014; Rüfenacht et al., 2016).

Besides the presence of planetary waves, signatures of atmospheric tides (ter-diurnal, semi-diurnal, diurnal) can be seen in middle atmospheric constituents or parameters like ozone, water vapor and temperature. Diurnal tides can be triggered by latent heat release within the troposphere (Hagan and Forbes, 2002) and can be of migrating or non-migrating nature. Overall complex interactions of atmospheric waves and coupling processes between different atmospheric layers exist. As Forbes (2009) assessed, the semi-diurnal solar thermal tide is a feature in the atmosphere of the earth, and serves to globally couple the troposphere, stratosphere, mesosphere, thermosphere and ionosphere.

Regarding the variability of the solar radiation, two major regimes, the 11-year and 27-day solar cycle, can affect the chemical composition of the middle atmosphere. While signatures of the 27-day solar rotation cycle were found in mesospheric OH and $H_2O$ (Shapiro et al., 2012; Lainer et al., 2016) and stratospheric $O_3$ and temperature (Ruzmaikin et al., 2007), the 11-year variability is a profound feature for instance in stratospheric ozone and temperature. Labitzke et al. (2002) performed general circulation model (GCM) simulations to study the impact of the 11-year solar cycle on geopotential height and temperature and made comparisons to observations.

Besides direct observations of middle atmospheric wind as a proxy for dynamical patterns, it is common to use observations of $H_2O$ or $O_3$ that can serve as diagnostic and dynamical tracers, even from ground-based profile measurements (Liu et al., 2013; Lainer et al., 2015), due to their relative long chemical lifetime, which is on the order of weeks (Brasseur and Solomon, 2006).

Here we report on ground-based observed water vapor oscillation in the mesosphere above Switzerland (46.88 °N, 7.46 °W) with a period of around 18 hours and investigate the temporal and monthly mean characteristics. This is to our knowledge the first study that explores a quasi 18-hour dominant wave mode in wintry (Northern Hemisphere) upper mesospheric conditions with passive microwave radiometry techniques.

Literature indicates that the 18-hour oscillation in mesospheric water vapor could be connected to the presence of low-frequency gravity-waves (GW), also called inertia-gravity waves, of similar apparent period. Li et al. (2007) described an 18-hour inertia-gravity wave. They used sodium-lidar measurements to probe the atmosphere between 80 and 110 km. In a 80-hour lasting campaign (December 2004) observations of temperature, sodium density, zonal and meridional wind were conducted. A linear least square data fitting revealed strong amplitudes in the wind fields with a characteristic increase with altitude. Wind amplitude peaks were detected between 96 and 101 km. The 18-hour signal was also present in temperature and sodium density, but less distinct. By applying linear wave theory, for details see e.g. Chapter 2 in Nappo (2002), an estimation of the horizontal wave propagation direction (245 °), wavelength (about 1800 km) and phase speed (28 m s$^{-1}$) could be determined for the first time with experimental data from a single instrument (Li et al., 2007).





We show with a much larger data set in which months the 18-hour or nearby period modes dominate in the water vapor profiles obtained by the Middle Atmospheric Water Vapor Radiometer (MIAWARA, Deuber et al. (2004)). The MIAWARA instrument has an upper measurement limit at approximately 75 km (0.02 hPa) and does not reach the same altitudes as a sodium-lidar system. The advantage of microwave radiometers is that they can measure during day and night and are not

critically influenced by the occurrence of clouds whereas lidar instruments usually are.

Gravity waves are a natural feature of a stably stratified atmosphere, where the squared Brunt-Väisälä frequency $N^2 > 0$. In general, GW can be classified into three types, with either low, medium or high intrinsic wave frequencies $\hat{\omega}$ (Fritts and Alexander, 2003). The role of atmospheric GW is to transport and deposit momentum by wave-breaking. Besides shear instability GW breaking events are an important source of turbulent kinetic energy production near the mesopause (Fritts et al.,

2003). As the sub-spectrum of gravity waves is large, plenty of different triggering mechanisms exist, including: Orographic lifting, spontaneous emission from jet streams and fronts, convective systems or water waves on oceans. Strong emissions of atmospheric gravity waves of low frequency (periods from a few hours to about 24 hours) were detected in the exit region of jets in the upper troposphere, as presented by Plougonven and Zhang (2014) and references therein. In the jet exit region the air flow is from north to south and establishes a force that decelerates the air when it leaves the jet streak (Shapiro and Keyser,

1990). The vertical motion resulting from the described mechanism leads to rising air in the north quadrant and sinking air in the south quadrant of the jet streak system. For an inertia-gravity wave gravity/buoyancy is not the only restoring force. The Coriolis force needs to be considered, since the horizontal wavelengths can reach up to 2000 km (Maekawa et al., 1984) with an estimated Rossby number $\mathrm{Ro} < 1$ of the flow system. A coherent 10.5 h low-frequency GW packet with vertical wavelengths between 4–10 km has been studied by Nicolls et al. (2010). They suggest a geostrophic adjustment of the tropospheric jet

stream a few days before the actual observation as the main triggering mechanism of the inertia-gravity wave packet.

In our study we use water vapor simulation data from the specified dynamics (SD) version of the Whole Atmosphere Community Climate Model (WACCM) for comparison with the ground-based observed water vapor oscillation. SD-WACCM wind and potential temperature fields are used to determine intrinsic period, horizontal propagation direction, horizontal and vertical wavelengths of the model resolved 18-hour inertia-gravity wave. As our research group at the Institute of Applied Physics

(University of Bern) operates unique microwave radiometers to observe mesospheric zonal and meridional wind components (Rüfenacht et al., 2014), we analyze wind measurements from October 2015 to March 2016 to investigate possible 18-hour wave events in the upper mesosphere.

In Sect. 2 the data sets from ground-based measurements, model simulations and Aura MLS satellite observations are described. The data processing methodology and the underlying numerical approach is part of Sect. 3. Section 4 describes and

analyses the results and some distinguished features of the present 18-hour wave oscillation (Sect. 4.2) and implications are addressed. Concluding remarks are provided in Sect. 5.





## 2   Data sets

In this section we describe the instruments and numerical models used to evaluate and analyze the quasi 18-hour wave activity. Two ground-based microwave radiometers (retrieving water vapor or wind), the NASA operated EOS (Earth Observing System) Aura MLS (Microwave Limb Sounder) instrument and data from SD-WACCM simulations are explained.

### 2.1   Ground-based microwave observations

The advantage of ground-based microwave radiometry is to continuously measure the amount of atmospheric trace gases at altitudes between roughly 30 and 80 km under most environmental conditions. Observations are possible during day, night and under cloudy conditions. The technique is widely used to study the middle atmosphere (Kämpfer et al., 2012). The Front-End of the MIAWARA instrument receives emissions from the pressure broadened rotational transition line of the $H_2O$ molecule. The center frequency of the transition line is 22.235 GHz. For studying oscillations with periods shorter than one day, a high temporal resolution of a few hours with an evenly spaced time series is required. In our case the MIAWARA water vapor time series has a temporal resolution of 3 hours. The $H_2O$ retrieval from 3 hourly integrated raw spectra is based on the optimal estimation method (OEM) as presented in Rodgers (2000). We use the ARTS/QPACK software (Eriksson et al., 2005, 2011), where the OEM is used to perform the inversion of the atmospheric radiative transfer model ARTS. The FFT (Fast-Fourier Transform) spectrometer at the Back-End of MIAWARA has a resolution of 60 kHz and the retrieval takes an overall spectrum bandwidth of 50 MHz. A monthly mean zonal mean Aura MLS climatology provides the a priori water vapor profile and additionally Aura MLS is used to set the pressure, temperature and geopotential height in the retrieval part. MIAWARA is part of NDACC (Network for the Detection of Atmospheric Composition Change) and is persistently probing middle atmospheric $H_2O$ from the Atmospheric Remote Sensing observatory in Zimmerwald (46.88 °N, 7.46 °E, 907 m a.s.l.) close to Bern since 2006. In the stratosphere the vertical resolution of the water vapor profiles is 11 km and degrades to about 14 km in the mesosphere (Deuber et al., 2005). A recent validation against the Aura MLS v4.2 water vapor product (Livesey et al., 2015) revealed that for most months and altitudes the relative differences between MIAWARA and Aura MLS are below 5 % (Lainer et al., 2016).

As our water vapor retrieval uses a fixed 3-hour integration, the measurement response (MR) in the mesosphere is sufficiently high during months from October to March/April. The MIAWARA $H_2O$ time series between October 2014 and March 2016 is shown in Fig. 1 and the MR criterion of 80 % is represented by the white horizontal lines. Except for some outliers we consider the upper measurement limit to range within 0.02–0.04 hPa during the NH winter season. In the summer season, when the humidity in the troposphere is high, our $H_2O$ retrieval from 3 hour integration time has a significant lower measurement response. It is not possible to get information that is sufficiently a priori independent above approximately 0.1 hPa in the upper mesosphere. We anticipate that all additional MIAWARA related plots will not show upper and lower measurement limits due to the fact that any a priori contribution cannot have an effect on a sub-diurnal variability as it is not resolved in the used seasonal climatological data. Further we note that we miss 1 week of MIAWARA data due to hardware problems beginning in the end of December 2015. This data gap pops up as white bar in the MIAWARA $H_2O$ time series.





Local instrument related temperatures at the MIAWARA measurement site, such as outdoor temperatures, indoor temperatures, mixer temperatures, Aquiris FFT FPGA (Field Programmable Gate Array) temperatures, hot-load and receiver temperatures, have been analyzed for oscillations that could have an influence on the observed wave signatures in the $H_2O$ retrieval data. The individual temperature amplitudes were examined for several months. Local peaks in the monthly mean amplitude

spectra occur close to 24 and 12 hours due to insolation. No distinct amplitude peaks have been found in the period range between 15 and 18 hours. Additionally the atmospheric opacity at 22.235 GHz obtained from tipping curve measurements and the measurement response in the water vapor retrieval were examined by the same filter algorithms for waves with periods between 6 to 30 hours. No correlations to the $H_2O$ wave spectra or prominent amplitude peaks between 15 and 21 hours were found during months when a distinct 18-hour $H_2O$ wave was existent in the MIAWARA data.

To our knowledge and made efforts, artificial effects leading to the observed 18-hour variability can be excluded and therefore the wave is expected to be of atmospheric origin. We aim to report on findings based on middle atmospheric observations and model simulations. Revealing possible sources of a 18-hour inertia-gravity wave is beyond the scope of this paper.

In 2012 the novel wind radiometer WIRA (Rüfenacht et al., 2014) has been developed at the Institute of Applied Physics at the University of Bern. It is the only instrument capable to steadily observe wind in the otherwise sparsely probed atmospheric

layer between 35 and 70 km altitude. Other techniques like rocket based meteorological measurements (Schmidlin, 1986) can provide data in this region but suffer from high operational costs, that makes them suitable for short campaigns but not for continuous observations. WIRA, a ground-based passive microwave heterodyne receiver, observes the Doppler shifts of the pressure-broadened emission line of ozone at 142 GHz. As for MIAWARA, the retrieval of zonal and meridional middle atmospheric wind components is based on OEM. The measurement uncertainty ranges from 10 to 20 m s$^{-1}$ and the vertical

resolution varies between 10 and 16 km. For more detailed information about the instrument we point the reader to papers by Rüfenacht et al. (2012, 2014). In order to resolve the 18-hour wave the retrieval was pushed to the limits by using measurements with an integration time of 6 hours only, instead of the usual 24-hour averages. Therefore, a new retrieval version which improves the wind accuracy of the mesospheric wind estimates has been used in this study. The WIRA instrument is capable to resolve the quasi 18-hour wave over Bern in the zonal wind vector component for only short time periods in the pressure

range 0.1–1 hPa. One of these time periods is between 2015-12-05 and 2015-12-09, and Fig. 2 shows the corresponding zonal wind profile time series. In the whole altitude domain the measurement response of the radiometer is greater than 0.8.

## 2.2   EOS Aura MLS satellite observations

Space-borne temperature observations are often used to investigate gravity wave activity (Ern et al., 2004; Hocke et al., 2016). With incorporating MLS temperature data in the study we strive to confirm and supplement the wave signatures in $H_2O$ and

wind profile data. Two case studies of the SD-WACCM resolved 18-hour inertia-gravity wave revealed vertical wavelengths below 6 km (Sect. 4.2.2). Therefore, temperature profiles from the Aura MLS satellite instrument are used to search for oscillations with vertical wavelengths smaller than 6 km. Oscillations in temperature profiles are helpful to identify local vertical structures associated with inertia-gravity waves. Increasing wave periods and horizontal wavelengths typically lead to decreas-



ing vertical wavelengths as it can be deduced from the dispersion relation for inertia-gravity waves (Fritts and Alexander, 2003).

For the location of Bern usually two temperature profiles per day are available in the Aura MLS data set. The vertical resolution in temperature profiles from MLS v4.2 is 4.5 km between 261 hPa and 100 hPa. It increases to 3.6 km at ∼30 hPa

and decreases to 4.3 km at 10 hPa. At higher altitudes the dropping in vertical resolution is ongoing and reaches values of 5.5 km (∼3 hPa), respectively 6 km (0.01 hPa) (Livesey et al., 2015). A vertical interpolation to a 1 km grid was applied to fulfill the Nyquist-Shannon sampling theorem. The along track resolution of the temperature measurements is about 165 km within the pressure range 261–0.1 hPa.

## 2.3  SD-WACCM model simulations

In our study the CESM (Community Earth System Model) v1.2.2 model version is used. It couples sub-models of the atmosphere, land, ocean and sea ice. The whole atmosphere model, called WACCM (Marsh et al., 2013) is a capacity of CAM (Community Atmosphere Model) v4 (Collins et al., 2006). Our simulations make use of the specified dynamics atmosphere model version SD-WACCM, whereby the model becomes nudged by 6-hourly GEOS5 (Goddard Earth Observing System 5) meteorological analysis data (horizontal winds, temperature, surface pressure, surface wind stress, latent and sensible heat

fluxes) every internal model time step interval of 30 minutes (Kunz et al., 2011; Lamarque et al., 2012). The nudging rate linearly decreases from 10 % below 50 km to 0 % between 50–60 km. A horizontal grid resolution of 1.9 ° (latitude) and 2.5 ° (longitude) is specified. Overall, 88 atmospheric layers up to 140 km with a resolution range between 0.5–4 km exist. The vertical coordinates are of hybrid sigma-pressure kind and follow the terrain near the surface whereas a transition to pure pressure coordinates takes place up to an altitude of approximately 17 km. The chemistry module of SD-WACCM is based on MOZART

(Model for OZone and Related chemical Tracers, Emmons et al. (2010)).

As described by Lin and Rood (1997), gravity waves are handled explicitly in a way consistent to the semi-Lagrangian advection scheme. A numerical scheme revealed by McFarlane (1987) is used to parameterize GWs triggered by orography. Gravity waves emerging from other sources and their propagation is implemented according to a scheme of Lindzen (1981), given that convection is the main initiation mechanism. The use of more parameterization schemes to cover non-resolved

25  GW sources can significantly improve the model performance within the stratosphere in regard of a better representation of observed dynamical variability (Limpasuvan et al., 2012).

We use the SD-WACCM simulation output for one grid point that represents our measurement location of Bern (7.4 °E, 47 °N). The following model variables are processed: Water vapor volume mixing ratio, potential temperature, zonal $u$ and meridional $v$ wind components (nudged). Every hour a data value is available on each vertical grid point. On the one hand

oscillations in SD-WACCM water vapor and wind, which have periods close to 18 hours, are investigated and on the other hand the potential temperature is used to compute the Brunt-Väisälä frequency $N$ (Eq. (1)) together with the perturbation time series $u'$, $v'$ to apply the hodograph method (Sawyer, 1961) in order to determine model resolved inertia-gravity wave





characteristics.

$$N = \sqrt{\frac{g}{\Theta} \frac{\partial \Theta}{\partial z}} \tag{1}$$

## 3   Numerical methods

In order to derive the wave spectrum in the MIAWARA and SD-WACCM data time series for Bern, we applied the following
numerical methods. A digital band-pass filter (non-recursive finite impulse response) with a comprised Hamming window is
applied to MIAWARA and SD-WACCM data to extract amplitudes of hidden oscillations of periods between 6 and 30 hours.
Performing windowing methods to measurement time series ensure that the data endpoints fit together and smooth out short-
term fluctuations to put longer-term cycles to foreground. Therefore the spectral leakage can be reduced (Harris, 1978). In
Studer et al. (2012) the numerical structure of the band-pass filter has been shown. Lately the filter has been used to investigate
the impact of the 27-day solar rotation cycle on mesospheric water vapor (Lainer et al., 2016) and to analyze the quasi 16-
day planetary wave during boreal winter (Scheiben et al., 2014). We follow the advice from Oppenheim et al. (1989) and
run the filter with a zero phase lag forward and backward along the measurement and simulation time series. The cut-off
frequencies of the passband attenuation are either set to 5 % or 16.6 % (depending on the analysis method) of the initialized
central frequency. The central frequency prearranges the size of the Hamming window which is the triple-fold of the central
period. Our filter and window setup guarantees a fast adaptability to data variations in time. In order to show how the 18-hour
$H_2O$ wave amplitudes appear in SD-WACCM compared to the MIAWARA water vapor analysis, a time dependent Pearson
product-moment correlation coefficient (PPMCC) is computed.

In general the application of hodograph method in an atmospheric environment where single monochromatic gravity waves
appear is valuable for obtaining intrinsic frequencies, propagation directions and as a result the horizontal wavelengths $\lambda_h$.
Quite large uncertainties can however occur by estimating $\lambda_h$, as shown by Zhang et al. (2004). But they note that for inertia-
gravity waves with short vertical wavelengths the uncertainties become smaller. A substantial number of gravity wave studies
(Li et al., 2007; Plougonven and Teitelbaum, 2003; Baumgarten et al., 2015) made use of the hodograph analysis.

A fitted hodograph ellipse from the wind perturbation amplitudes ($u'$, $v'$) will rotate clockwise with increasing altitude for
the Northern Hemisphere (Tsuda et al., 1990), pursuant to the polarization relation (Eq. (2)), if the analyzed inertia-gravity
wave has a phase velocity tending downwards.

$$u' = \left( \frac{i\hat{\omega}k - fl}{i\hat{\omega}l - fk} \right) v' \tag{2}$$

The wave phase propagation has a direction given by the vector $(k, l, m)$. The intrinsic frequency $\hat{\omega} = \omega - k\bar{u} - l\bar{v}$ is the
frequency that would be observed while moving with the background wind $(\bar{u}, \bar{v})$ (Fritts and Alexander, 2003). The intrinsic
frequency can be obtained from the hodograph ellipse by calculating the ratio between the major and minor axis $\varepsilon$ according
to $\hat{\omega} = \varepsilon \cdot f$, where $f$ is the inertial frequency. Further, the inertial period can be easily calculated with $T_f = 2\pi f^{-1}$. The
GW direction of horizontal propagation $\varphi$ is determined by the orientation of the major ellipse axis. For now, two possible





propagation directions exist. Tsuda et al. (1990) show how an explicit solution can be identified. The relation between the vertical and zonal wind perturbation velocities

$$\frac{w'}{u'} = -\frac{k}{m} \tag{3}$$

where $k$ and $m$ are the wave numbers in horizontal and vertical direction, can be used to clarify the GW horizontal propagation orientation. Since $k$ is positive and $m$ negative if the GW propagation consists of a downward phase velocity, the vertical wind perturbation of the gravity wave has to be upward. As soon as the GW propagation direction $\varphi$ and intrinsic frequency $\hat{\omega}$ are known, Doppler relation and the dispersion relation for inertia-gravity waves can be applied to calculate the vertical and horizontal GW wavelengths $\lambda_z$ and $\lambda_h$ (e.g., Fritts and Alexander, 2003; Liu and Meriwether, 2004).

In this context it is important to mention that an apparent wave frequency $\omega$ is larger than the intrinsic frequency $\hat{\omega}$ if the wave is moving in the direction of the air flow. It can be expressed with the Doppler relation ($\hat{\omega} = \omega - 2\pi\lambda_h^{-1}U_k^{-1}$), where $U_k$ is the horizontal wind speed in direction of wave propagation. In simplified terms, the Coriolis dependent dispersion relation can be wrapped up as

$$\hat{\omega}^2 = \frac{N^2 \cdot \lambda_z^2}{\lambda_h^2} + f^2 \tag{4}$$

## 4 Results

### 4.1 Monthly mean wave spectra

A mean wave amplitude is obtained by averaging amplitude series over time. For example, a final $H_2O$ wave amplitude spectrum as presented in Fig. 3 and 4 is created by computing the monthly-averaged amplitudes as a function of the period. The period range goes from 6 to 30 hours with a resolution of 1 hour. Overall, 12 months of microwave radiometric water vapor measurements were processed. The mean amplitude wave spectra reveal that except for October 2014 the highest wave amplitudes are located in the 18-hour period band for the 2014/15 period. During October 2014 a different regime close to 12 hour is dominating. Below 1 hPa amplitudes in water vapor are small. Regarding the 18-hour variability the altitude domain above 0.1 hPa is most interesting. During the 2015/16 period clear 18-hour signals can be found in November 2015, January and February 2016 (Fig. 4). During the other 3 months (October and December 2015, March 2016) high amplitudes show up with periods near 12 and 24 hours (tidal patterns). Clear and high wave amplitudes at exactly 18 hours are found in December 2014 (Fig. 3c), January 2016 (Fig. 4d) and February 2016 (Fig. 4e). The altitude region, where the 18-hour oscillation is prominent, is mostly above 0.1 hPa. We find monthly mean quasi 18-hour $H_2O$ amplitudes in the range 0.25–0.35 ppm. Prominent wave events with sharp 18-hour periods happened in January and February 2016. In the following the focus is put on these events. Monthly mean spectra as derived from SD-WACCM are presented in Fig. 5. The SD-WACCM simulation data show no clear amplitude maximum in the 15-21 hr period band as MIAWARA observations do. Nevertheless amplitudes of 0.15 to 0.25 ppm are present above 0.1 hPa. For the analysis of the SD-WACCM $H_2O$ mean wave spectrum a sampling rate of 2 hours was applied. Within the subsequent section we investigate how often the 18-hour wave packets have been observed in the MIAWARA water vapor time series. Comparisons to SD-WACCM results are given.



### 4.2 Quasi 18-hour wave characteristics

We present the whole temporal evolution (12 months) of the water vapor oscillations in the quasi 18-hour period band for MIAWARA and SD-WACCM as comparison. For all the comparisons with the numerical simulations we have to be aware of the fact that parameterized gravity-wave effects in models are generally considered poorly constrained by observations (Fritts
and Alexander, 2003).

For January 2016 we perform two hodograph analysis with SD-WACCM to show exemplary how the quasi 18-hour wave characteristics is resolved in the numerical model. Since WIRA has not been capable to derive meridional winds with high quality from October 2015 to March 2016 in Bern, it cannot be used jointly with SD-WACCM for the hodograph method. Instead, we concentrate on the temporal 18-hour wave amplitude in zonal wind in the case of WIRA data between 5th and 9th
of December 2015.

### 4.2.1 Temporal behavior

Both the absolute and the relative amplitudes of the 18-hour wave in the MIAWARA observations show that this wave occurs quite regularly during the investigated winter months (Figs. 6 and 8). The amplitudes of the wave become highest above the mid-mesosphere (0.1 hPa). This is in quite good agreement with the SD-WACCM analysis (Figs. 7 and 9). Local (in time)
amplitudes reach up to 0.5 ppm or 12 % in relative units. Scheiben et al. (2013) showed that in the altitude range from 3 hPa to 0.05 hPa the diurnal $H_2O$ amplitudes do not exceed 0.05 ppm. Any kind of interference of migrating tides is not expected to produce water vapor amplitudes as large as observed for the quasi 18-hour wave. A comparative high difference by a factor of ten is present between MIAWARA diurnal and quasi 18-hour wave amplitudes. The $H_2O$ wave emergence takes place in packets, what is reminiscent of inertia-gravity waves. A growing of 18-hour wave amplitudes with decreasing pressure is
observed and likewise supports an involvement of inertia-gravity waves. Correlation coefficients of vertical amplitude profiles of the quasi 18-hour wave between MIAWARA and SD-WACCM are given in Fig. 10. Regarding the time period over 12 months in total, the correlations can reach up to about 0.8 with 95 % confidence.

From October 2015 to March 2016 WIRA has observed middle-atmospheric wind over Bern. The 6 hourly binned WIRA data often show larger gaps of a few days in the time series at the altitudes of interest. This makes it difficult to search for
18-hour wave activity, when continuous measurements are necessary. During the highly dynamic phase at the beginning of December 2015 WIRA data with the required quality are available to complement the water vapor data. Continuous WIRA observations between 5th to 9th of December 2015 reveal a strong zonal 18-hour wave component (Fig. 11b). At the 5th and 8th of December the band-pass filtered absolute wave amplitudes reach values between 40 and 50 m s$^{-1}$. From 7th towards the 9th of December the zonal wind as observed by WIRA (Fig. 2) is significantly accelerating. Between 0.3–0.7 hPa horizontal
wind speed patterns almost double.

The observation of high wave amplitudes do not reach pressure levels below 0.3 hPa during the first event (5th of December). The second event (8th of December) shows high amplitudes almost covering the whole WIRA altitude range from 0.1–1 hPa. By comparing the WIRA quasi 18-hour wave amplitude time series to SD-WACCM results (Fig. 11c) we find similar activity



patterns. However, there are also significant differences. Most strikingly the SD-WACCM 18-hour zonal wind amplitudes are by a factor of 2 lower than that of WIRA. The second event is not represented below 0.2 hPa. However, the vertical differences between observations and model might to some part also be due to the limited vertical resolution of the instrument smearing out features in altitude.

In the MIAWARA water vapor data the same wave events on 5th and 8th of December reach 18-hour oscillation amplitudes of 0.35 ppm and 0.45 ppm. For the wave event on 8th of December 2015, the pressure level of the amplitude maximum (0.1 hPa) is in agreement between MIAWARA and SD-WACCM. For WIRA the maximum is located a bit lower in altitude (at 0.25 hPa). The $H_2O$ maximum for the first event is located at higher altitudes than for WIRA and SD-WACCM winds. The temporal extension and behavior appears to agree better with WIRA than with SD-WACCM.

An interesting period occurred in January 2016. A distinct 18-hour water vapor amplitude at 0.04 hPa is found, that is higher (by 0.1 ppm) than the diurnal amplitude signature at 0.4 hPa (Fig. 4d). In particular the focus is put on 10 days between 2016-01-16 and 2016-01-26. Figure 12 presents the 18-hour band-pass filtered amplitude time series of MIAWARA $H_2O$ and SD-WACCM zonal and meridional wind vector components. The water vapor amplitudes are highest in the pressure range 0.01–0.1 hPa and reach about 0.4 ppm (Fig. 12a). During the presented 10 days, 3 periods of higher wave activity are detected:

2016-01-16 12 UTC to 2016-01-18 00 UTC, 2016-01-20 00 UTC to 2016-01-21 00 UTC and 2016-01-22 12 UTC to 2016-01-25 12 UTC. The computed horizontal wind component wave amplitudes of the nudged model simulations (Fig. 12b and c) reach up to 20 m s$^{-1}$, but time of the amplitude maximums are different to those of $H_2O$. In the case of the 2016-01-20 12 UTC maximum in $H_2O$ the zonal wind amplitude maximum occurs 12 hours later roughly at the same altitude (0.03 hPa). For 2016-01-20 00 UTC and 2016-01-20 12 UTC we perform the propagation analysis (black vertical lines in Fig. 12) as described in

Sect. 3.

### 4.2.2  Propagation analysis

The quasi 18-hour inertia-gravity wave, represented by the SD-WACCM model wind perturbations, is characterized by a hodograph analysis on 2016-01-20 00 UTC and 12 hours thereafter (Fig. 13a and b). During these times the 18-hour oscillation appears in the MIAWARA water vapor data. The clockwise rotation of the $u'/v'$ hodograph with rising altitude means that the

wave is propagating upwards and its phase progression is downwards.

The horizontal wave propagation on 20th January 00 UTC is heading towards southwest at 230 ° and veers westward to 260 ° 12 hours later. The unambiguous direction along the main ellipse axis is determined by the vertical wind perturbation profiles (Fig. 13c and d) associated to the quasi 18-hour oscillation (Tsuda et al., 1990). The horizontal wavelengths $\lambda_h$ are estimated from the Doppler relation to be about 1884 km, respectively 1385 km. Intrinsic wave periods are close to 14 hours.These SD-

WACCM model related findings are in general agreement to observational results of an 18-hour oscillation with horizontal wavelengths of 1800 km in the altitude range 82–103 km (Li et al., 2007). A similar direction of wave propagation is likewise present.

The mean Brunt Väisälä frequency $N$ (0.02 s$^{-1}$) is determined by the potential temperature profiles and is used together with $U_k$, the wind speed projected onto the $k$-direction of the wave, to estimate the associated vertical wavelengths $\lambda_z$. For





the presented case in Fig. 13a (b), the mean $U_k$ value between 0.01–0.3 hPa is $8\,\mathrm{m\,s^{-1}}$ ($-6\,\mathrm{m\,s^{-1}}$) which leads to vertical wavelengths of roughly 5.8 and 4.4 km. Persistent inertia-gravity wave signatures in the middle atmosphere above northern Norway were studied by Baumgarten et al. (2015) and apparent vertical wavelengths between 5–10 km occurred. The given information about an intrinsic period of an identified monochromatic wave is 9.4 hours, which is 4 hours shorter than for our shown example.

Common vertical wavelength of less than 6 km of the 18-hour wave initialized the idea to analyze Aura MLS vertical temperature profiles with respect to oscillations with similar wavelengths. One intention is the identification of potential sources of the quasi 18-hour oscillation in middle atmospheric water vapor. In the following Sect. 4.3 a closer look at vertical middle atmospheric temperature anomalies over Bern is given.

## 4.3 Aura MLS temperature profiles

Aura MLS v4.2 temperature limb sounding profiles close to the location of Bern are high-pass filtered in vertical direction. The objective is to see whether effects of low frequency inertia GW activity can be detected and whether they are in agreement with our previous given results. The key figure we present here is the development of temperature amplitudes $\Delta T_{HF}$ between 30 and 80 km from high-pass filtered temperature profiles at a cut-off vertical wavelength of 6 km. Since 3–6 km is the vertical resolution limit of MLS in the middle atmosphere, shorter wavelengths cannot be resolved. A Doppler wind and temperature lidar measurement campaign by Baumgarten et al. (2015) identified a number of inertia-gravity wave cases at altitudes between 60–70 km with emphasis on upward propagation and $\lambda_z$ in the range 5–10 km. One observed GW had an apparent period of approximately 11 hours. For higher GW periods, e.g. 18 hours from our reported results, even smaller vertical wavelength have to be expected and we decided to take $\lambda_z < 6\,\mathrm{km}$ as the high-pass cut-off criterion.

The analysis of Aura MLS temperature data makes use of two measured temperature profiles per day in proximity to the MIAWARA water vapor observation site. We show the two time periods as for the temporal investigation of the 18-hour wave activity (Sect. 4.2). The two covered 6 months periods in 2014/2015 and 2015/2016 are included. Figure 14 shows vertical as well as temporal temperature structures in the atmosphere induced by low frequency gravity waves. Wavelike disturbances in time occasionally occur below 50 km showing higher temperature amplitudes up to 1–2 K, for instance between 2014-12-27 and 2015-01-16 or the week around 2015-12-22. A prominent feature for both time periods is a band-like structure between roughly 60–70 km (0.2–0.04 hPa) with more or less consistently high $\Delta T_{HF}$ amplitudes. Pronounced quasi 18-hour MIAWARA water vapor wave activity is found within or close to the observed band-like structure of large temperature oscillations.

The wave propagation beyond an altitude of 70 km seems to be suppressed for longer time periods. Irregularly and during rather short periods waves are able to travel further upwards. The time range 2015-12-18 to 2015-12-26 is an example when low frequency gravity waves with $\lambda_z < 6\,\mathrm{km}$ were observed throughout the middle atmosphere (Fig. 14b), since wave dissipation and reflections were suppressed.

The vertical propagation is very sensitive to the actual atmospheric background wind conditions. As Charney and Drazin (1961) show, the reflection of planetary waves due to wind shear in the middle atmosphere is a prevailing atmospheric process.



Further an environment of high wind shear favors the elimination of a fraction from any present atmospheric gravity wave spectrum (Hines and Reddy, 1967).

Broadly speaking, a suppression of wave dissipation and reflection processes can be linked to wind shear properties in different atmospheric layers. One commonly used variable to reflect speed and directional shear is the bulk shear vector that

can be obtained from the conventionally used $u/v$ hodograph. The bulk shear vector is figured out by the subtraction of the two relative to ground wind vectors at the the lower and upper boundary of the considered atmospheric shear layer.

Table 1 summarizes the monthly mean magnitude of the bulk shear vector. The layer between 64 and 74 km has the highest average amount of bulk shear. In the considered time period, the values for February 2015 and 2016 reach its peak with around $74 \, \mathrm{m \, s^{-1}}$ ($65 \, \mathrm{m \, s^{-1}}$). These high wind shear conditions are likely to be the reason why the GW activity above 70 km was

low (Fig. 14). Since the wind shear has a high temporal variability, occasionally low shear values occur and allow waves to travel higher up. Within the 54–64 km layer much lower bulk shear is found mostly between 20 and $30 \, \mathrm{m \, s^{-1}}$, which makes additional parameters necessary to explain the sharp boundary in temperature amplitudes right below 60 km. The location of the stratopause with the accompanied local temperature maximum at around 50 km altitude could have a significant effect on the propagation of waves.

## 5   Concluding remarks

For the first time a dominant 18-hour wave in mesospheric water vapor has been reported from ground-based measurements. A unique data set from the MIAWARA instrument with a temporal resolution of 3 hours has been examined for wave signatures with periods between 6–30 hours. Two winter time periods were used to present monthly mean wave spectra of $H_2O$. For a considerable number of months prominent wave signatures in the quasi 18-hour (15–21 hours) period band have been identified.

The packet-like occurrence in time and growing amplitudes with decreasing pressure are in agreement with inertia-gravity wave characteristics. Whether the observed wave is a direct image of a low frequency gravity wave is not definitely clear. A clear physical connection between the temporal coherence of 18-hour water vapor and wind amplitudes has not been found. But individual 18-hour $H_2O$ amplitude profiles of MIAWARA and SD-WACCM regularly reach high correlation coefficients of more than 0.6 or less frequently even more than 0.8.

It has been shown that the WIRA instrument is capable to resolve sub-diurnal oscillations, although a larger continuous observation time for such studies would be desirable. The quality of the WIRA meridional wind component measurements have still a potential for improvement and could contribute to wave characteristic analyses also in regard of validations to models. In this study only a few days of WIRA measurements could be used to compare with MIAWARA and SD-WACCM. The MIAWARA $H_2O$, SD-WACCM and WIRA zonal wind wave patterns in December 2015 have shown temporal correlation

with mismatch in altitude. The January 2016 case study did neither exhibit an obvious temporal nor vertical correlation between observed MIAWARA $H_2O$ and SD-WACCM wind 18-hour wave signatures. By now we do not have a comprehensive understanding about the links between the observed quasi 18-hour wave in $H_2O$ and wind, although water vapor should mostly be dynamically controlled in the mesosphere and sub-diurnal time range. Photochemistry and vertical transport have small





impact on mesospheric water vapor leading to a mean $H_2O$ lifetime on the order of weeks (Brasseur and Solomon, 2006). We conclude that more sophisticated numerical model studies are needed, which go beyond the scope of this paper and our possibilities, to explain the found circumstance.

What has to be considered by comparing the appearance of wind and water vapor wave amplitudes is that the latter may only be observed where the vertical or horizontal gradient in the volume mixing ratio is high. This might lead to high differences and low qualitative correlations in time and space. It should be added that vertical and horizontal advection affect the $H_2O$ variability. Further, as mentioned before, parameterized gravity wave effects in models are generally considered poorly constrained by observations (Fritts and Alexander, 2003).

With single instrument observations of all necessary variables, such in Li et al. (2007) with a sodium-lidar, the identification of inertia-gravity wave characteristics is less interdependent. The disadvantage of single instrument observations is the possibility of measurement artifact spreading over all measured parameters. Independent observations of atmospheric wave patterns by different instruments can reduce misinterpretations.

Although the elliptical fit in our hodograph perturbation data is not perfect with the SD-WACCM model data, we show that meaningful inertia-gravity wave parameters can be identified: Intrinsic periods of about 14 hours, south-westward to westward propagation direction with downward phase progression, horizontal wavelengths of 1884 km and 1385 km and vertical wavelengths of less than 6 km. These results can foster future studies such as model validations to high quality observations from ground by for example radar, lidar or microwave instruments.

*Author contributions.* ML was responsible for the ground-based water vapor measurements, performed the data analysis and prepared the manuscript. KH designed the filter algorithm and contributed to the interpretation of the results. RR is in charge of WIRA, the ground based wind radiometer, and provided wind retrieval data. FS carried out SD-WACCM simulations. NK is the lead of the project group. All authors read and approved the current manuscript version and declare that they have no conflict of interest.

*Acknowledgements.* This work is supported by Swiss National Science Foundation Grant 200020-160048 and MeteoSwiss in the frame of the GAW project "Fundamental GAW parameters measured by microwave radiometry". Rolf Rüfenacht is supported by Post-doc Grant P2BEP2-165383. We acknowledge NASA for access to Aura MLS data and the NCAR CESM working group for providing the SD-WACCM model code.



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





**Table 1.** Monthly mean magnitude of bulk shear vector $[\mathrm{m\,s^{-1}}]$ calculated from SD-WACCM model output within four different middle atmospheric layers of 10 km depth. The time period from October 2014 to March 2015 (upper panel) and October 2015 to March 2016 (lower panel) is covered.

| Month / Shear layer | Oct 2014 | Nov 2014 | Dec 2014 | Jan 2015 | Feb 2015 | Mar 2015 |
|---|---|---|---|---|---|---|
| 64–74 km | 33.2 | 59.7 | 43.0 | 43.9 | 73.7 | 55.7 |
| 54–64 km | 24.5 | 28.5 | 34.7 | 28.9 | 26.4 | 26.7 |
| 44–54 km | 13.0 | 30.8 | 28.6 | 36.5 | 29.8 | 18.0 |
| 34–44 km | 16.8 | 40.5 | 31.2 | 27.4 | 45.9 | 28.7 |

| Month / Shear layer | Oct 2015 | Nov 2015 | Dec 2015 | Jan 2016 | Feb 2016 | Mar 2016 |
|---|---|---|---|---|---|---|
| 64–74 km | 38.6 | 52.1 | 59.7 | 59.2 | 65.1 | 36.5 |
| 54–64 km | 20.4 | 28.9 | 31.2 | 41.3 | 29.6 | 31.0 |
| 44–54 km | 14.1 | 21.6 | 23.5 | 23.7 | 15.3 | 16.3 |
| 34–44 km | 17.2 | 35.6 | 43.8 | 20.2 | 36.0 | 21.7 |

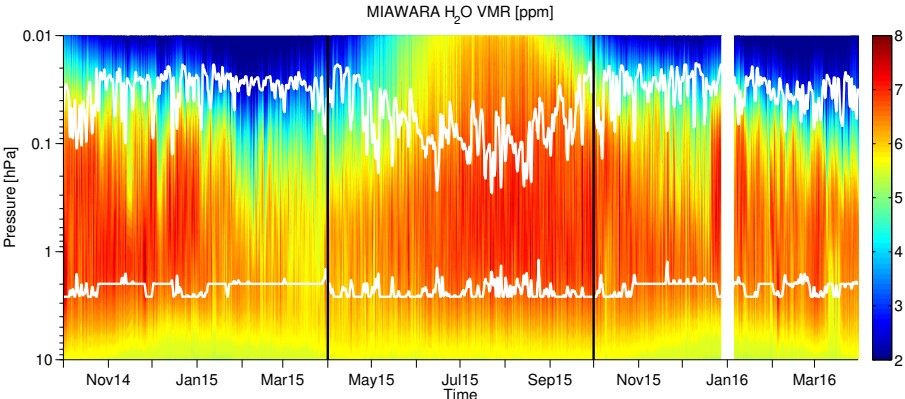

**Figure 1.** The water vapor volume mixing ratio [ppm] time series measured by MIAWARA between October 2014 and March 2016. The horizontal white lines indicate at which pressure levels the measurement response drops below 80 %. During the more humid and warm season between April and September 2015 the data will not be used. This is marked by the vertical black lines. A measurement gap occurred between 2015-12-28 and 2016-01-04 as shown by the white bar.





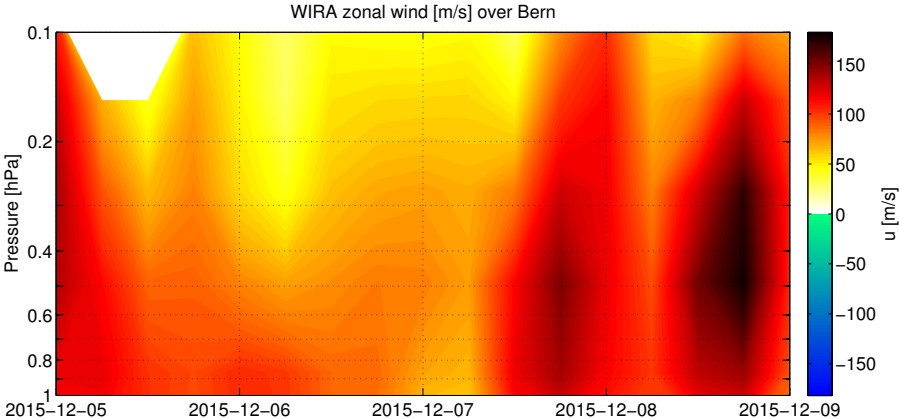

**Figure 2.** The zonal wind vector component time series measured by WIRA between 2015-12-05 and 2015-12-09 in the pressure range 0.1–1 hPa.

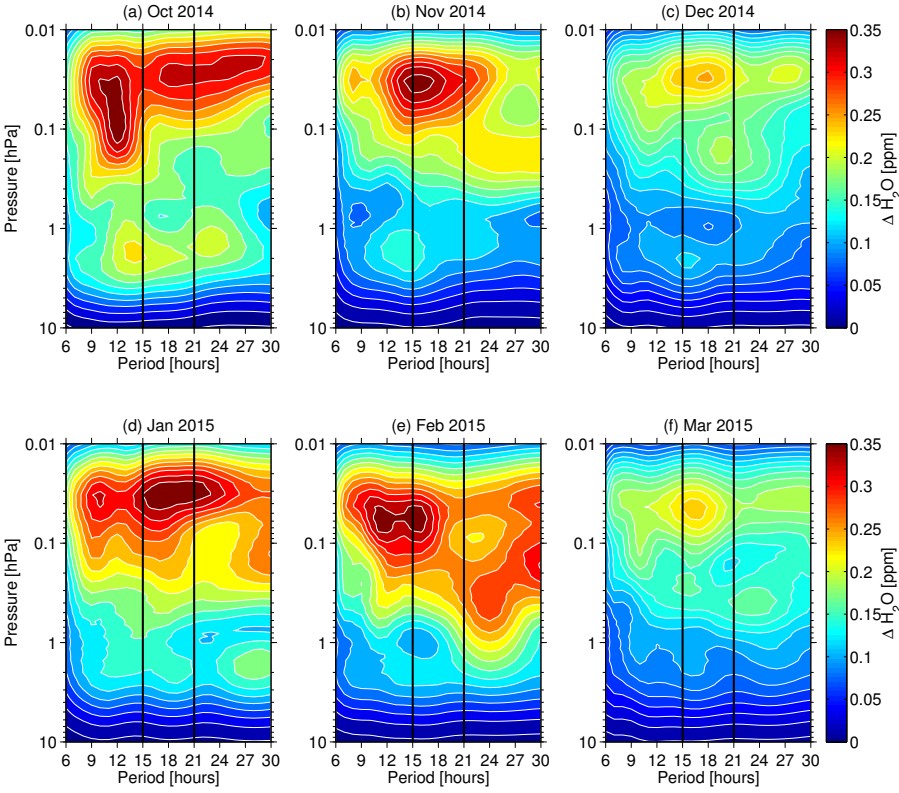

**Figure 3.** The MIAWARA water vapor monthly mean wave spectrum with periods between 6 and 30 hours. Shown is the result of the $H_2O$ amplitudes [ppm] for the months October 2014 to March 2015 (a–f). The border of the quasi 18-hour period band (15–21 hours) is indicated by the vertical black line pair.





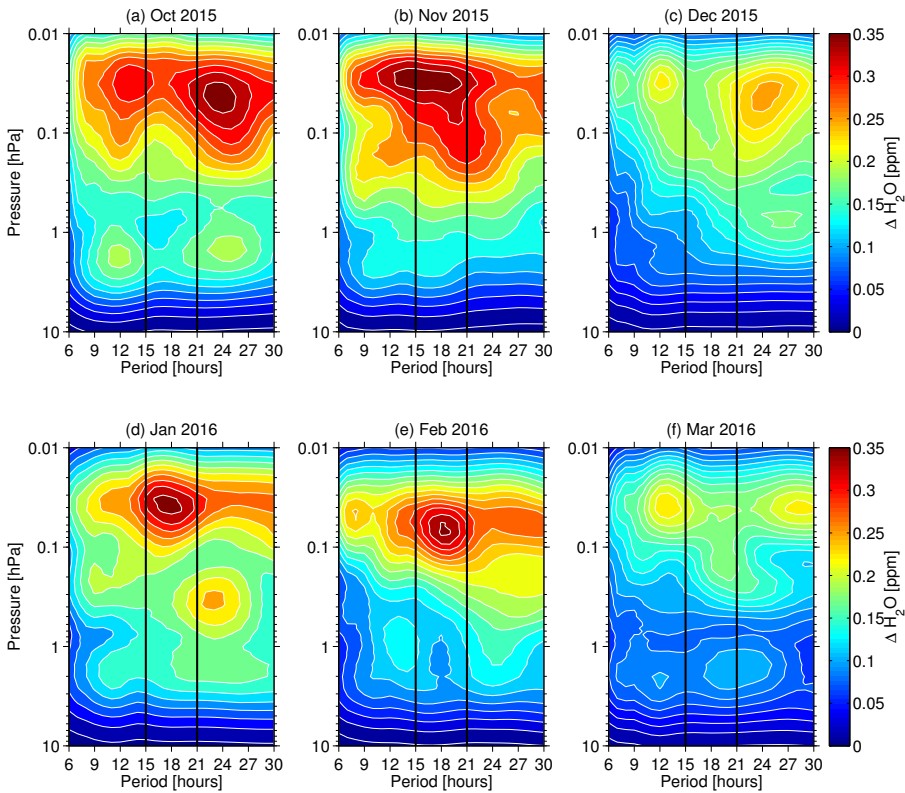

**Figure 4.** Same as Fig. 3, but here we focus on the months October 2015 to March 2016 (a–f).

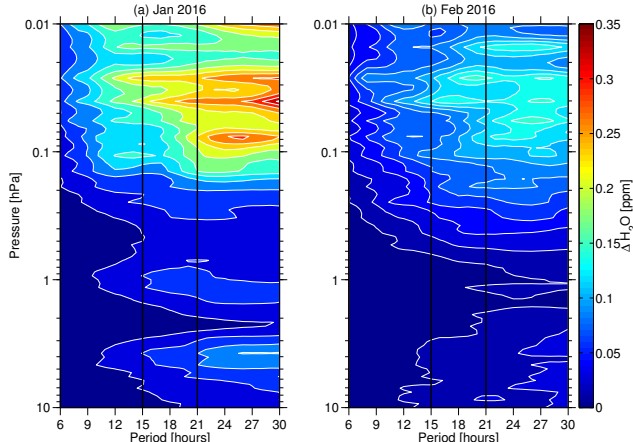

**Figure 5.** The SD-WACCM water vapor monthly mean wave spectrum with periods between 6 and 30 hours. Shown is the result of the H$_2$O amplitudes [ppm] for the months January (a) and February (b) 2016. The border of the quasi 18-hour period band (15–21 hours) is indicated by the vertical black line pair.



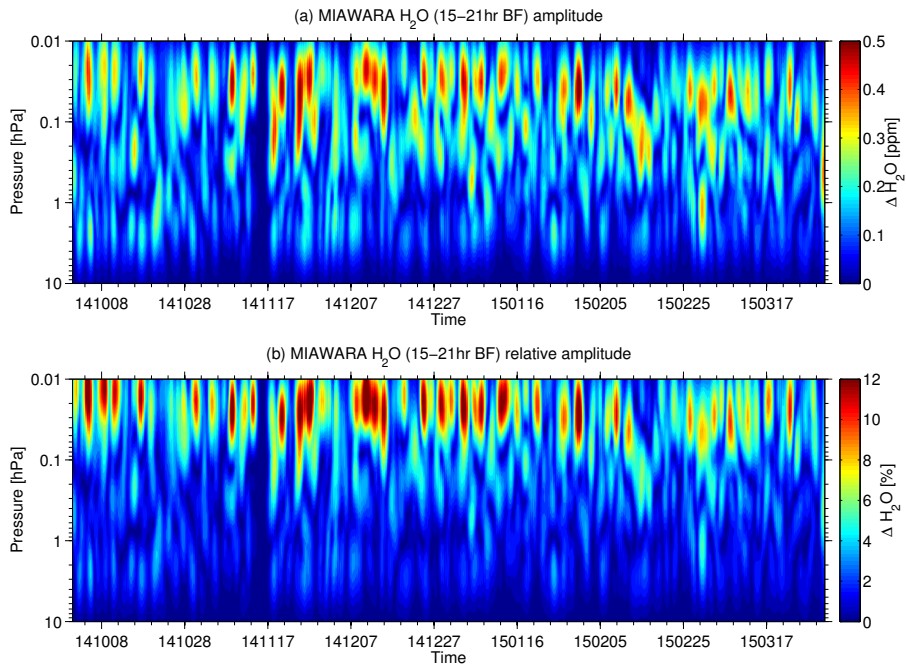

**Figure 6.** Temporal evolution of wave amplitudes derived from band-pass hamming-window filtered MIAWARA $H_2O$ VMR time series with cut-off periods at 15 and 21 hours. Shown is the time period from October 2014 to March 2015.



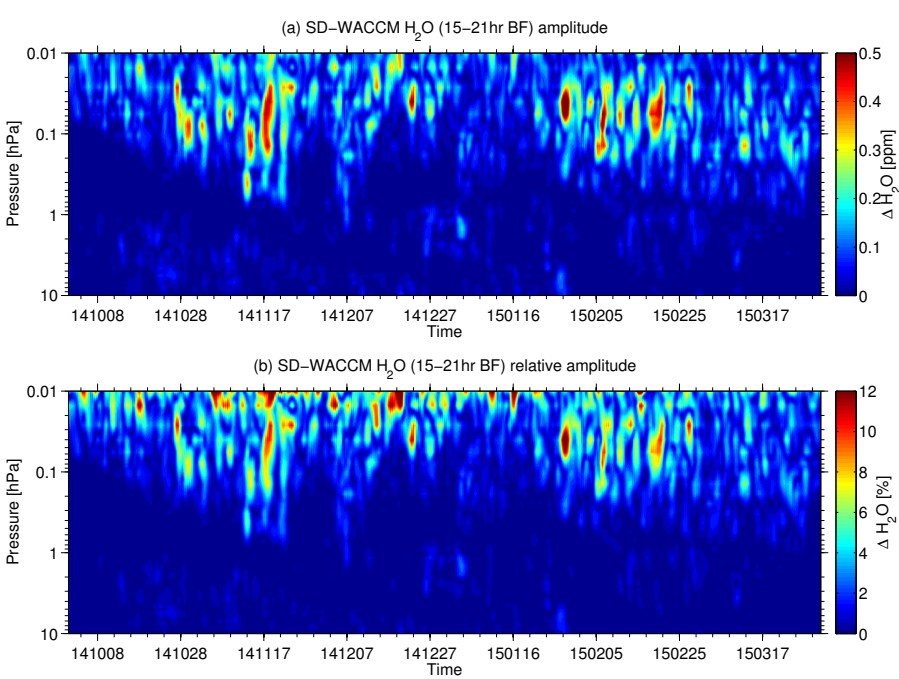

**Figure 7.** Temporal evolution of wave amplitudes derived from band-pass hamming-window filtered SD-WACCM $H_2O$ VMR time series with cut-off periods at 15 and 21 hours. Shown is the location of Bern and the time period from October 2014 to March 2015.





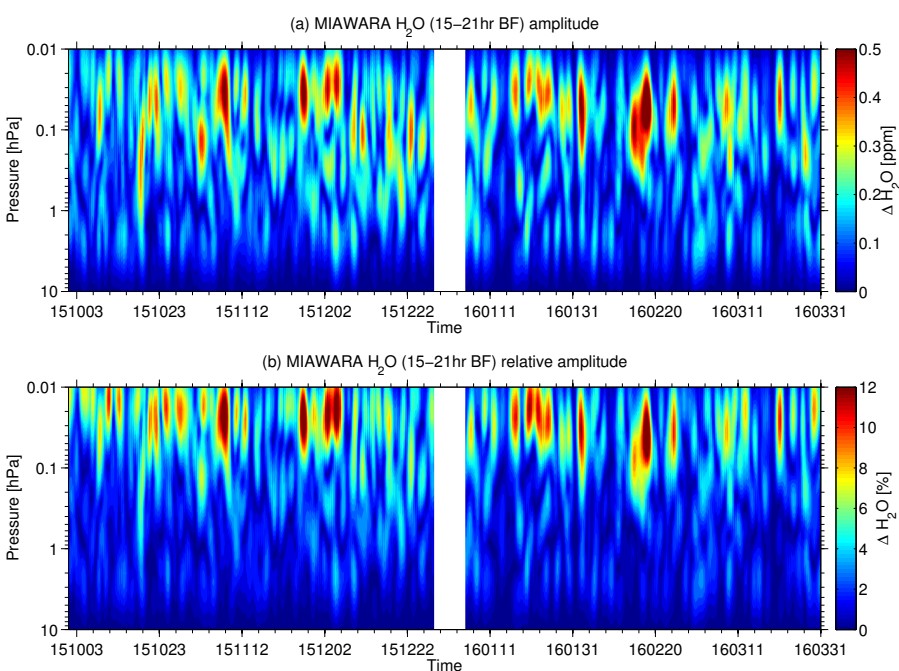

**Figure 8.** Same as Fig. 6, but here the time period from October 2015 to March 2016 is shown. The measurement gap between 2015-12-28 and 2016-01-04 is indicated by the white bar.





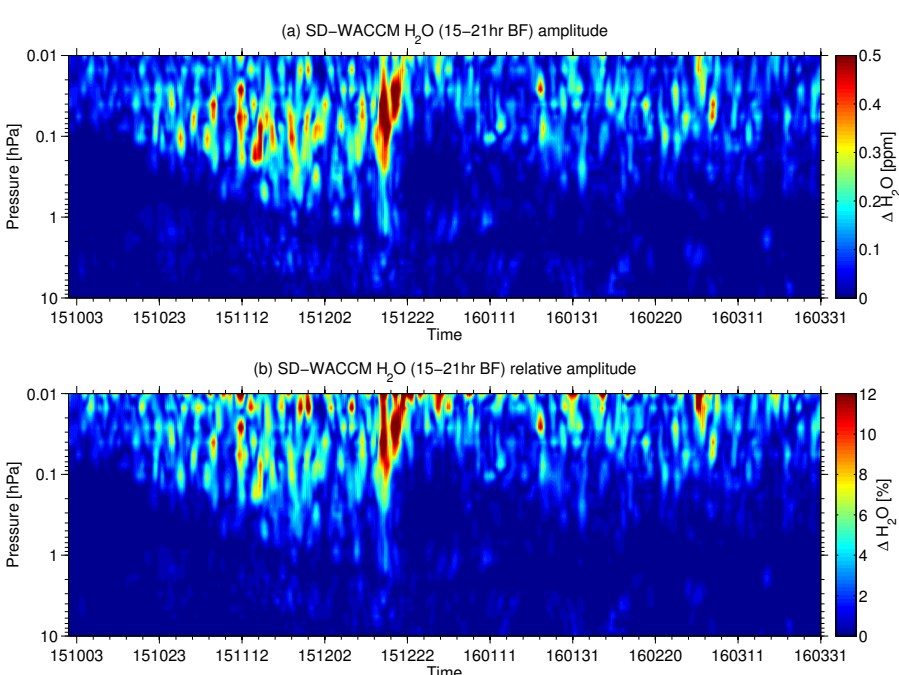

**Figure 9.** Same as Fig. 7, but here the time period from October 2015 to March 2016 is shown.





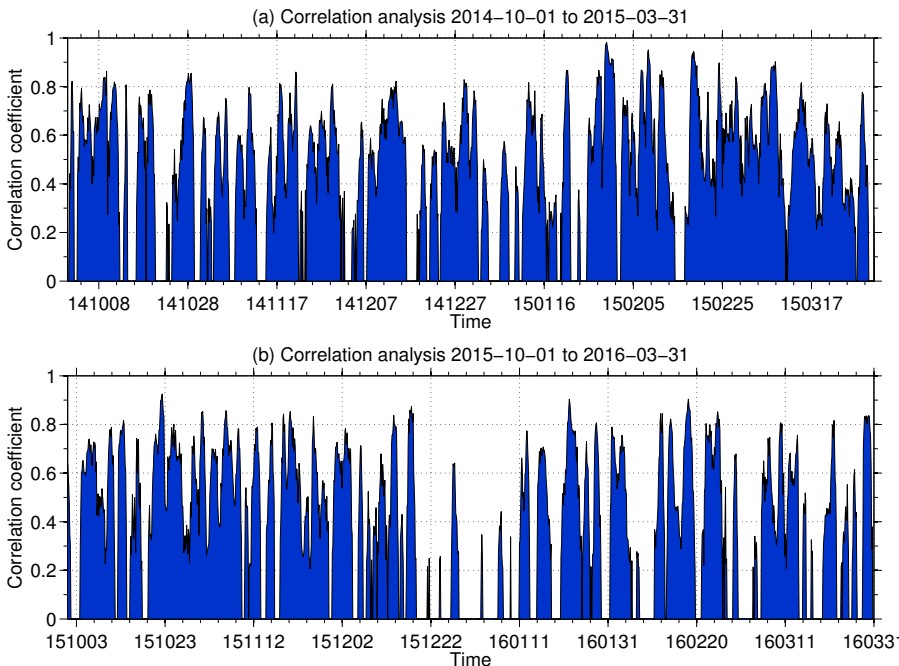

**Figure 10.** Filled area plot of Pearson product-moment correlation coefficient between MIAWARA and SD-WACCM quasi 18-hour (15–21 hr) wave amplitude profile time series (in blue). In (a) the time period from 2014-10-01 to 2015-03-31 and in (b) from 2015-10-01 to 2016-03-31 is considered. Here only significant correlation coefficients (confidence interval >95 %) are plotted.





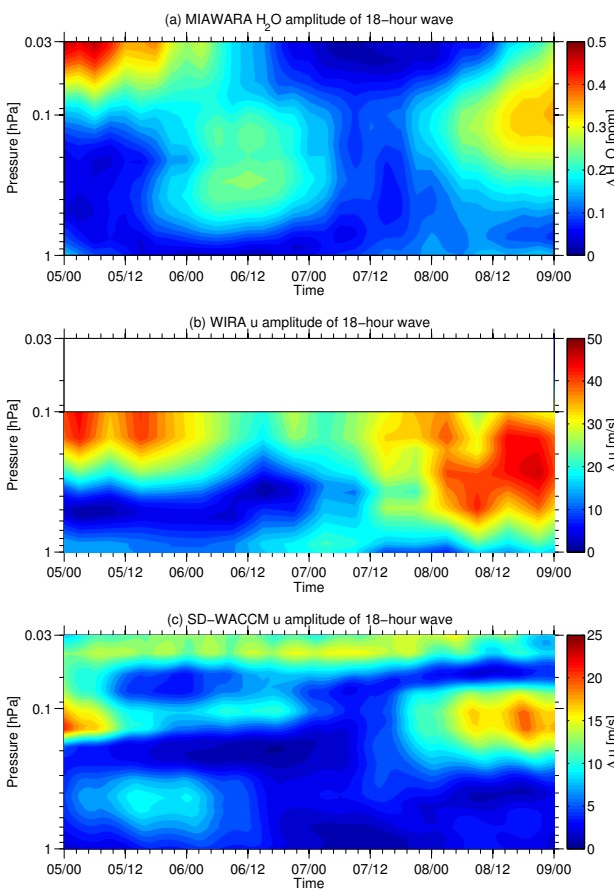

**Figure 11.** 18-hour band-pass filtered absolute wave amplitudes in the pressure range 0.03–1 hPa between 2015-12-05 and 2015-12-09. Upper panel (a) shows water vapor amplitudes as observed by MIAWARA, middle panel (b) shows zonal wind amplitudes as observed by WIRA and the bottom panel shows the zonal wind amplitudes from SD-WACCM model simulations.





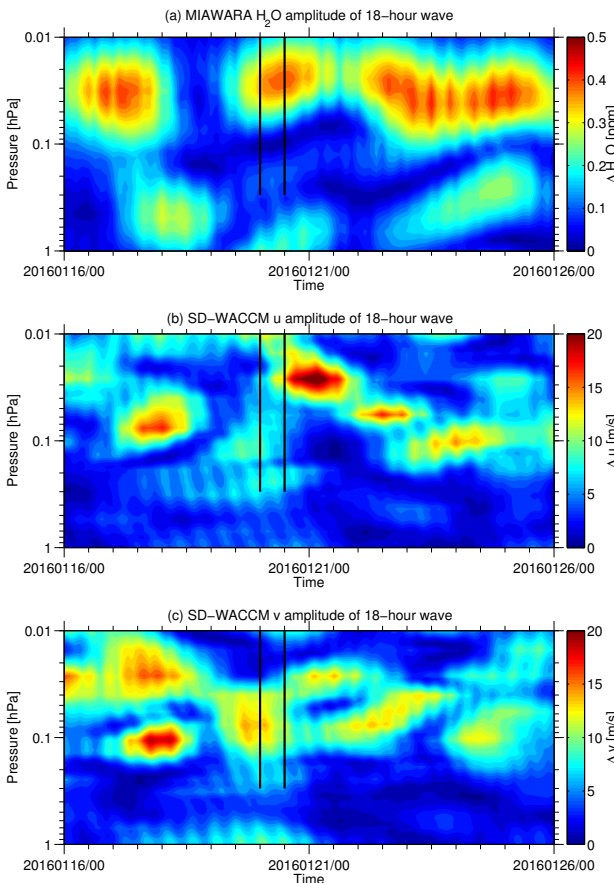

**Figure 12.** 18-hour band-pass absolute wave amplitudes in the pressure range 0.01–1 hPa between 2016-01-16 and 2016-01-26. Upper panel (a) shows water vapor amplitudes as observed by MIAWARA, middle panel (b) shows zonal and the bottom panel (c) meridional wind amplitudes from SD-WACCM model simulations. The vertical black lines mark the dates and pressure ranges of the exemplary hodograph analyzes (Fig. 13).





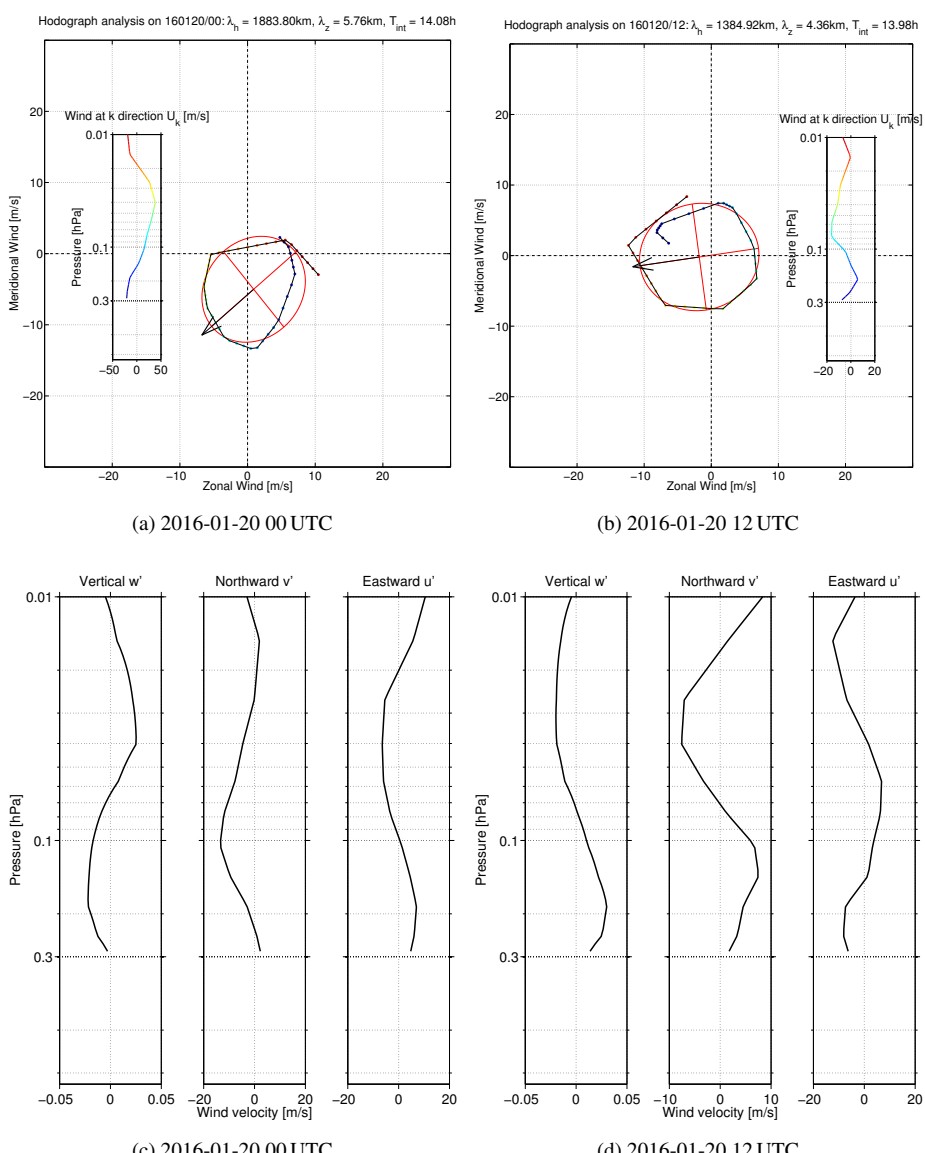

**Figure 13.** Upper panels (a,b) show the hodograph of the 18-hour SD-WACCM zonal and meridional wind perturbations. An elliptical fit (red) is applied to the data points. Wave propagation direction $k$ is indicated by the black arrow. Embedded in the hodograph figures, are the vertical profiles of the background wind speed profiles projected onto the $k$-direction. The used color scheme separates different pressure levels, from blue to red means increasing altitude (decreasing pressure). Lower panels (c,d) show accordant to the overlying hodographs profiles of the vertical, eastward and northward 18-hour wind perturbations.




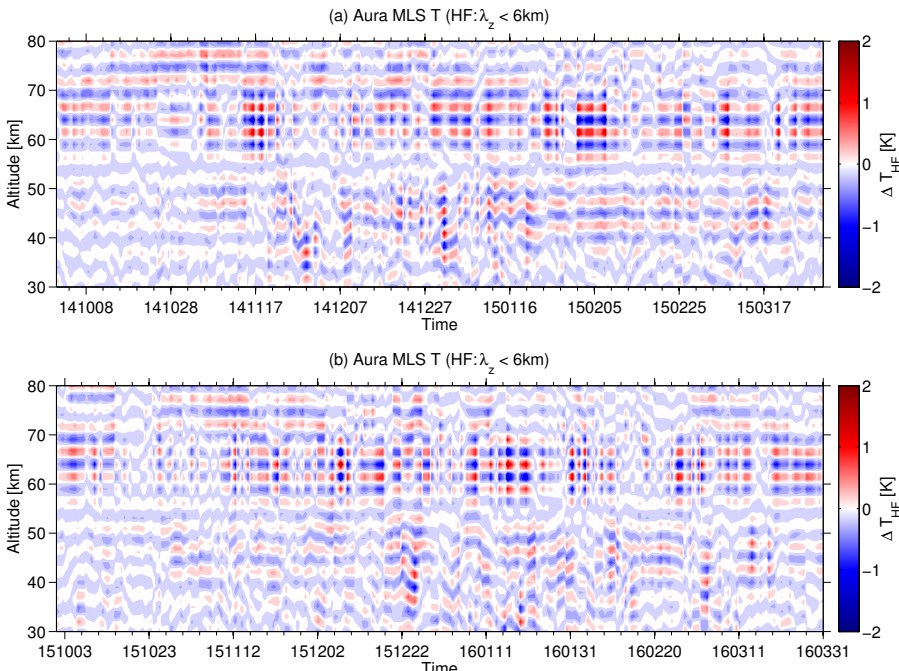

**Figure 14.** Vertical high-pass filtered Aura MLS temperature profiles from 2014-10-01 to 2015-03-31 (a) and from 2015-10-01 to 2016-03-31 (b). High frequencies in the temperature profiles are passed through, i.e. with vertical wavelengths $\lambda_z < 6\,\mathrm{km}$. The plot shows the filtered temperature oscillations $\Delta T_{HF}$ in the altitude range 30–80 km.