# Peer review of "Quasi 18-hour wave activity in ground-based observed mesospheric H2O over Bern, Switzerland"

_Atmospheric Chemistry and Physics, 2016_

## Referee Comment (RC1) · Anonymous Referee #2 · 5 Feb 2017

The authors analyse gravity waves based on microwave water vapour observations in the mesosphere. They also investigate mesospheric winds based on microwave radiometer observations, and analyse WACCM GCM model results and EOS Aura satellite data. They find that waves with ~18 hr period are frequently seen in the mesosphere. The topic is suitable for ACP. In particular, the microwave observations are of interest to the community and the paper deserves publication in principle.

However, the analysis of wave parameters from the WACCM model and Aura data is partly unclear and to a certain degree questionable. The authors use model and observations with vertical resolution of several km, so I do not see how waves with wavelengths < 6 km should be resolved. At least major modification is required.

Major issues

[Figure]

WACCM model results: a resolution of 2.5° means that only waves with wavelength > 400 km or so are resolved. So this is certainly a different part of the spectrum than observed with the microwave radiometer. Figs. 4 and 5 show that the spectra are completely different, and the only commonality is the height range, where the waves maximise. But this is only a similarity and one cannot really identify common waves in the model and observations.

The hodograph analysis in Fig. 13 requires explanation. There is some theory given in section 3, but it is not well described what the authors really did to obtain the wave parameters. Obtaining the intrinsic frequency from ïĄě and then using the Doppler relation to get the horizontal wavelength? How was the observed frequency defined, from the radiometer measurements? And what is the error of this analysis? If the intrinsic frequency and horizontal wavelength is known, the dispersion relation will give the vertical wavelength, but from Fig. 13 a vertical scale of some 20 km is visible, is it possible that the difference 20 km vs. < 6km comes from uncertainties of the analysis? ïĄě is close to unity, and then a relatively small error might give a large relative error for the wavelength. WACCM cannot resolve waves with short vertical wavelength. The authors refer to Baumgarten et al. (2015), but in their wind and temperature residuals the short wavelength is immediately visible.

Aura/EOS observations: The vertical resolution is less than 3 km, so I do not see how waves with wavelengths < 6 km can be resolved. The description of Fig. 14 is not very clear. I assume that it shows temperature residual profiles every 12 hr? In the mesosphere, Fig. 14 shows maxima/minima constantly at the same level. This does not look like a real atmospheric phenomenon, and it rather seems as if these are the original data levels and the waves seen are due to aliasing. Analysis of Aura data therefore must be explained in much more detail, and possible effects of resolution have to be discussed. I doubt, however, that the results in Fig. 14 really show the gravity waves.

Minor issues

P 2, introduction, l 13: The paragraph on the solar effects may be deleted. At least regarding the 11-year cycle, as the paper deals with gravity waves and not long-term, interannual variability.

P3, L7: maybe replace "frequency by "angular frequency", at least when first introduced.

P9, L20, Fig. 10: How was the correlation calculated? For each profile separately, so that the correlation is strong if the amplitudes maximise at the same height? This would not mean too much, in particular would not give information on whether the amplitudes appear simultaneously or not. If the correlation is insignificant, is it then simply set to zero?

P11, L 13: "temperature amplitudes", do you mean "residuals" or filtered temperatures as in Fig 14?

P11, L 21: how do you know that it is the 18 hr wave that is analysed from the temperature profiles?

P 11, L 23: Which kind of temporal structures? Long-period variations of the waves?

Fig 13, caption: what means "background wind speed?"

---

## Referee Comment (RC2) · Anonymous Referee #3 · 4 May 2017

Review of Lainer et al

I should first note that I have deliberately not read the other review. So if my comments seem in any way duplicative, consider them to be independent opinions.

Major comment

Their result seems interesting, is broadly plausible and certainly suitable for ACP. My main question, and it's a serious one, is how can they actually observe this kind of oscillation in their H2O data? Any wave in a conserved tracer field should only be manifest if the vertical gradient of the tracer is small enough relative to the vertical displacement. The oscillations seen in their data (i.e. Figure 1) seem very large for what is supposedly only a 6 km vertical wavelength. And further, their vertical resolution seems insufficient to capture a 6 km wave. I am comfortable with a limb viewer such

as MLS seeing this wave, but a vertical sounder seems less likely. On page 4, line 20, they give their vertical resolution as 11-14 km, but then say the 18 hour wave has 6 km vertical wavelength. If that is the case, then how can they see it? At a minimum, they need to present some simulations showing this. For example, the gravity wave community has spent considerable time and effort illustrating how waves are seen differently in limb vs. nadir sounders. Here, some sort of test case with an idealized wave is called for in order to be truly convincing, in my opinion. Or perhaps taking the WACCM fields and convolve them with the microwave averaging kernels. I'm worried that something else with a period of 18 hours is contaminating their retrieval and thus they are not actually seeing H2O oscillations. The authors jump too quickly to spectral analysis without first presenting more raw data and showing how it varies. The same question applies to their wind data which has stated resolution of 10-16 km

Minor comments

1. What is their integration time? On page 4, line 24, they say 3 hours. On page 5, line 22 they say 6.

2. If their measurement is only valid to 0.02 hpa (e.g. page 3), then they should cut off their plots at that level (e.g. Figures 1, 3, 6)

Grammar. In general, while their English is order of magnitudes better than my German, the grammar should be improved. An incomplete list follows.

End of abstract and beginning of Intro: They use "manifold" in adjacent sentences which seems awkward.

Page 2, line 2: "Latter analyzed"... ?

Page 2, line 23 – either "a" or use plural

Line 1 on page 3- what is this supposed to mean?

Line 24-25 on page 4: again, a poorly expressed thought: I think I understand why

the winter data are more usable- due to lower tropospheric humidity. But this sentence implies something else. Do they mean that the measurement response in winter is sufficiently high that they can use a time integration as short as 3 hours (as opposed to say, a day?). If so they need to express that more clearly.

Page 5,line 10: "To our knowledge and made efforts," ? "made efforts" is poor English.

Page 12, line 25 "is capable of resolving"

---

## Referee Comment (RC3) · Anonymous Referee #4 · 22 May 2017

The submitted paper mainly deals with observations of ground based mesospheric water vapour using the MIAWARA experiment near Bern (46.88° N) during 12 winter months from Oct 2014 – March 2015 and from Oct 2015 –March 2016. The authors found in 7 of these 12 months dominating oscillations with periods between 15 and 21 h. To interpret these oscillations, they used

a) additional nudged SD –WACCM simulations,

b) for case studies zonal winds from the Doppler wind radiometer WIRA at the same location, and

c) temperature profiles derived from Aura MLS satellite observations.

The combination of these datasets itself is valuable and suitable for publication in ACP.

However the structure and the conclusions of this paper requires substantial improvements.

From the reviewer point the highlights of the paper are:

(a) The high quality of continuous MIAWARA water vapour measurements (Fig 1).

(b) The observed monthly mean oscillations above 0.1 hPa during the winter months (Figs 3 and 4), but see also remark 4.

(c) The similarity and significant correlation between the bandpass filtered wave amplitudes derived from MIAWARA and the corresponding water mixing ratio derived from SD WACCM simulations (Figs 6 - 10) see remark 5.

(d) The case study from 5 – 9 Dec 2015 using MIAWARA (water vapour), WIRA (u), and SD WACCM (u) (Fig 11), see remark 6.

(e) The case study from 16 -26 Jan 2016 using MIAWARA (water vapour), and SD WACCM (u and v) (Fig 12), see remark 7.

General Remarks

1) Whereas the authors wrote in the introduction at page 5 lines 10 -12 "To our knowledge and made efforts, artificial effects leading to the observed 18-hour variability can be excluded and therefore the wave is expected to be of atmospheric origin. We aim to report on findings based on middle atmospheric observations and model simulations. Revealing possible sources of an 18-hour inertia-gravity wave is beyond the scope of this paper." However, in the following they are only focussing on an 18-hour inertia-gravity wave based on a single case study (Figs. 12 and 13). From the reviewer point, this generalization on all events is not valid because the difference between the observed period ($\sim$18h) and the inertial period of 16.4 h as upper limit for the intrinsic period at the latitude of Bern (46.88°N) requires more or less at least constant background winds to get the Doppler shift of the intrinsic GW frequencies which has not been shown here.

2) An oscillation with a period of about 18h can also be the result of a nonlinear wave-wave interaction of two waves, e.g. between quasi two day wave and the semidiurnal tide or between the semidiurnal and terdiurnal tide. This must be checked and considered as a possible reason for the obtained oscillations.

3) In contrast to Figs 3 and 4, the SD WACCM spectra show diurnal tidal waves with a poor spectral resolution, but no dominant oscillations between 15 and 21 h. It is surprising that there is such a similarity and significant correlation between the bandpass filtered wave amplitudes derived from MIAWARA and the corresponding water mixing ratio derived from SD WACCM simulations (Figs 6 - 10). Can you comment this?

4) Please explain and/or improve the spectral resolution presented in Figs 3-5.

5) Please define the term "relative amplitudes" as used in Figs 6 – 10.

6) The case study (d) from 5 - 9 Dec 2015 (Fig 11) shows similar wave amplitudes between MIAWARA (water vapour), WIRA (u), and SD WACCM (u) and gives confidence that, with meridional winds from WIRA, a better wave estimation at the same location will be possible. In the frame of the used title focussing on 18h waves, however, Fig 4 c show during this period only tides (12h, 24h) but nothing between 15 – 21 h

7) The hodographs in Figs 13 and the derived possible characteristics of a monochromatic gravity wave are based on band pass filtered model simulations. It is not clear for the reviewer, how realistic are these simulated amplitudes, where the gravity waves are handled consistent a parametrization (see Page 6, lines 21 – 26). The cited papers of Baumgarten et al. (2015) and Li et al. (2007) used LIDAR data with a high resolution to estimate their hodographs. Please consider also a Stokes parameter analysis to get a more averaged GW description instead of the "snapshot" hodograph of a single monochromatic wave. Furthermore it is recommended to add the dispersion and Doppler equation to the wave parameter estimation to improve the readability.

8) The AURA MLS temperatures and water vapour profiles are important for the MI-

AWARA data as described in Sec 2.1 (page 4). However, at altitudes of about 0.1 hPa , where the observed 18h oscillations have their maxima, the vertical resolution lies between deltah 5.5 and 6 km (see page 6, line 6) , so that only waves with vertical wavelengths larger than 2 x deltah (11 – 12 km) can be resolved. From this point, the filtered temperature profiles with vertical wavelengths < 6 km are questionable, at least above 0.1 hPa.

Technical corrections

9) Page 2 line 8 please add wind

10) Page 7 line 13 bandpass

11) Page 8 line 32 are given in the next Section.

---

## Author Comment (AC1) · 4 Jul 2017

**Quasi 18-hour wave activity in ground-based observed mesospheric water vapor over Bern, Switzerland**

*Martin Lainer, Klemens Hocke, Rolf Rüfenacht, and Niklaus Kämpfer*
* * *
**Final Response on ACPD paper acp-2016-1050:**
* * *
* * *
**Color Code:** Referee comments, Authors response, Link to relevant changes in manuscript
* * *
We would like to thank all anonymous referees for their efforts and comments which helped us to launch a revised manuscript version. This new version only focuses on the observations (mesospheric water vapor and wind) and the robustness of the analyzed data. The parts dealing with SD-WACCM model simulations and Aura MLS data are removed with regard to the reviewer comments. In consequence many comments will be answered only briefly due to the omission of manuscript parts.

Please find our point by point response to all three reviews below. A marked-up manuscript version is provided in the end.

**1 Response to Referee #2**

**Major issues**

(1)
WACCM model results: a resolution of 2.5° means that only waves with wavelength >400 km or so are resolved. So this is certainly a different part of the spectrum than observed with the microwave radiometer. Figs. 4 and 5 show that the spectra are completely different, and the only commonality is the height range, where the waves maximise. But this is only a similarity and one cannot really identify common waves in the model and observations.

We agree that the analysis of atmospheric wave parameters from the SD-WACCM model simulations and Aura MLS data was not sophisticated enough. As stated in the introduction, all parts in the manuscript dealing with WACCM or Aura MLS data are omitted now. This includes also the hodograph analysis, which was only performed with SD-WACCM data because the quality of the meridional wind observations by the Doppler wind radiometer WIRA was not good enough.

Section 2.3 is deleted. In Sect. 2 we now only describe the microwave radiometers and the corresponding data sets. In Sect. 4 the focus is put on the monthly mean $H_2O$

spectra (Sect. 4.1) and the temporal evolution of the 18-hour wave amplitudes in the $H_2O$ and zonal wind data (Sect. 4.2). All parts and figures mentioning and showing SD-WACCM results are removed (Figs. 5, 7, 9, 10, 11c, 12). In Section 4, a new subsection 4.3 is included and discusses the obtained results in regard of inertia-gravity waves or the possibility of a non-linear wave-wave interaction between the quasi 2-day wave and the diurnal tide.

(2)
The hodograph analysis in Fig. 13 requires explanation. There is some theory given in section 3, but it is not well described what the authors really did to obtain the wave parameters. Obtaining the intrinsic frequency from $\epsilon$ (ratio between the major and minor axis of hodograph ellipse) and then using the Doppler relation to get the horizontal wavelength? How was the observed frequency defined, from the radiometer measurements? And what is the error of this analysis? If the intrinsic frequency and horizontal wavelength is known, the dispersion relation will give the vertical wavelength, but from Fig. 13 a vertical scale of some 20 km is visible, is it possible that the difference 20 km vs. <6 km comes from uncertainties of the analysis? $\epsilon$ is close to unity, and then a relatively small error might give a large relative error for the wavelength. WACCM cannot resolve waves with short vertical wavelength. The authors refer to Baumgarten et al. (2015), but in their wind and temperature residuals the short wavelength is immediately visible.

With our instruments alone we were not able to make use of the hodograph method. In principle it requires a higher vertical resolution than our microwave instruments can provide. Since the SD-WACCM part of the manuscript is deleted, it does not make sense any more to present the hodograph method and the simulated results.

Section 4.2.2 (Propagation analysis) is omitted together with Figs. 12 and 13. The manuscript part about the numerical methods is much shorter now and only explains the spectral data analysis we use.

(3)
Aura/EOS observations: The vertical resolution is less than 3 km, so I do not see how waves with wavelengths <6 km can be resolved. The description of Fig. 14 is not very clear. I assume that it shows temperature residual profiles every 12 hr? In the mesosphere, Fig. 14 shows maxima/minima constantly at the same level. This does not look like a real atmospheric phenomenon, and it rather seems as if these are the original data levels and the waves seen are due to aliasing. Analysis of Aura data therefore must be explained in much more detail, and possible effects of resolution have to be discussed. I doubt, however, that the results in Fig. 14 really show the gravity waves.

Our Aura MLS analysis is to a certain degree critical. If the vertical wavelength is less than 6 km the Nyquist sampling theorem is not valid for the vertical resolution of the MLS satellite data. However the result of a 6 km vertical wavelength is based on model data and might not describe the real atmospheric situation. But with our observations

we were not able to derive the vertical or horizontal wavelengths parameters. Based on the results of the SD-WACCM hodograph analysis, we agree that a MLS temperature profile analysis is more or less pointless.

Sections about Aura MLS and the temperature profiles (Sect. 2.2 and 4.3) is deleted, including Fig. 14.

**Minor issues**

(1)
P 2, introduction, l 13: The paragraph on the solar effects may be deleted. At least regarding the 11-year cycle, as the paper deals with gravity waves and not long-term, interannual variability.

We agree that the paragraph on the solar variability misses the point.

Therefore, the paragraph on solar variability is deleted in the introduction.

(2)
P3, L7: maybe replace frequency by angular frequency, at least when first introduced.

We agree, that it is more correct to use the term angular frequency.

In the discussion Section 4.3 we use the term angular frequency when first introduced.

(3)
P9, L20, Fig. 10: How was the correlation calculated? For each profile separately, so that the correlation is strong if the amplitudes maximise at the same height? This would not mean too much, in particular would not give information on whether the amplitudes appear simultaneously or not. If the correlation is insignificant, is it then simply set to zero?

It is correct that the correlation was calculated for each profile separately and set to zero if it was not significant (95 % confidence). However in the new manuscript version no comparison to SD-WACCM is performed any more.

Figure 10 is not presented any more in the results.

(4)
P11, L 13: temperature amplitudes, do you mean residuals or filtered temperatures as in Fig 14?

With temperature amplitudes we referred to as filtered temperatures. But no Aura MLS temperature data is shown in the revised version of the manuscript now.

Section 4.3 and Figure 14 is not part of the manuscript any more.

(5)
P11, L 21: how do you know that it is the 18 hr wave that is analysed from the temperature profiles?

It is correct that we cannot say that the high-pass filtered time series of the temperature profiles is only related to the 18 hour wave. Only the SD-WACCM hodograph analysis suggested that the inertia-gravity waves have vertical wavelengths below 6 km and that value was used as upper limit in the filter settings.

We do not show any MLS data now.

(6)
P 11, L 23: Which kind of temporal structures? Long-period variations of the waves?

Needless to answer, since MLS data is not shown any more.

Same as before. Section 4.3 is deleted.

(7)
Fig 13, caption: what means background wind speed?

The background wind speed is the projected true wind speed in the direction of wave propagation (obtained from the hodograph analysis). The background wind speed as shown in Figure 13 is not a 18-hour filtered component of u and v. Since the comparison between SD-WACCM and WIRA / MIAWARA data is questionable we do not apply the hodograph analysis in the new manuscript version.

Figure 13 and the presented results are omitted now.

**2 Response to Referee #3**

**Major comment**

Their result seems interesting, is broadly plausible and certainly suitable for ACP. My main question, and it's a serious one, is how can they actually observe this kind of oscillation in their $H_2O$ data? Any wave in a conserved tracer field should only be manifest if the vertical gradient of the tracer is small enough relative to the vertical displacement. The oscillations seen in their data (i.e. Figure 1) seem very large for what is supposedly only a 6 km vertical wavelength. And further, their vertical resolution seems insufficient to capture a 6 km wave. I am comfortable with a limb viewer such as MLS seeing this wave, but a vertical sounder seems less likely. On page 4, line 20, they give their vertical resolution as 11–14 km, but then say the 18 hour wave has 6 km vertical wavelength. If that is the case, then how can they see it? At a minimum, they need to present some simulations showing this. For example, the gravity wave community has spent considerable time and effort illustrating how waves are seen differently in limb vs. nadir sounders. Here, some sort of test case with an idealized wave is called for in order to be truly convincing, in my opinion. Or perhaps taking the WACCM fields and convolve them with the microwave averaging kernels. I'm worried that something else with a period of 18 hours is contaminating their retrieval and thus they are not actually seeing $H_2O$ oscillations. The authors jump too quickly to spectral analysis without first presenting more raw data and showing how it varies. The same question applies to their wind data which has stated resolution of 10–16 km.

I guess with your statement "this kind of wave" you mean an inertia-gravity wave with a vertical wavelength below 6 km. You are right we would not be able to see such a wave. Have in mind that the 6 km wavelength was derived from the WACCM model. In consequence the 18-hour waves seen in WACCM and our water vapor or wind data are not comparable. From this point of view we even do not know if we see the effect of a inertia-gravity wave, which should then have a much larger vertical wavelength of at least ~20 km. We decided to completely remove the WACCM data analysis and with it the propagation analysis of the hypothetical model resolved 18-hour inertia-gravity wave. The focus is now only on our ground-based observations. Unfortunately we have not the expertise on simulating inertia-gravity waves. Instead of our model based simulation to explain our wave observations we try to show that our retrievals are robust and not contaminated by possible 18-hour oscillations of instrument related parameters such as measurement response or various temperatures (outdoor, indoor, mixer, FPGA, Hot-Load, receiver).
We will also include more raw data. In case of MIAWARA water vapor we show monthly time series averaged between 0.02–0.1 hPa, which shows how the amount of $H_2O$ varies in the altitude region where the 18-hour oscillation is observed. In case of the zonal wind measured by WIRA, we now show all observations which are available in December 2015. Since the whole SD-WACCM analysis is removed from the manuscript, a convolution by the microwave averaging kernels is redundant now.

We included new Figs. 2 and 3 showing monthly time series of MIAWARA $H_2O$ averaged between 0.02–0.1 hPa. Fig. 4 shows now a longer zonal wind profile time series from the WIRA radiometer (2. Dec to 15 Dec.). Still the spectral analysis can only be performed between 5. Dec to 9. Dec., so the WIRA plot in Fig. 9 has not changed. In order to show that instrument related temperatures do not have a dominant 18-hour oscillation mode we exemplary show monthly mean wave spectra of 6 temperature time series for January, February and March 2016 (new Fig. 10). The results are presented

in the beginning of the new Section 4.3 (Discussion). The spectral analysis in the new Fig. 11 shows dominating oscillations in the a priori contribution (respectively measurement response) of the water vapor retrievals. We have not identified any prominent oscillations in the defined quasi 18-hour period band. Thus we conclude that we indeed observe real atmospheric wave oscillations in our data sets. Of course still it is not clear what causes these oscillations in the $H_2O$ tracer field and zonal wind. Section 4.3 continuous to discuss about the inertia-gravity wave theory, but also on non-linear wave-wave interactions with regard to other published studies (eg. Li et al. (2007); Nicolls et al. (2010); Lieberman et al. (2017)

**Minor comments**

(1)
What is their integration time? On page 4, line 24, they say 3 hours. On page 5, line 22 they say 6.

On page 4, line 24 we were describing the MIAWARA water vapor retrieval and this uses an integration time of 3 hours. On the next page 5 line 22 we talk about the wind radiometer WIRA, which uses an integration time of 6 hours. Due to the omission of SD-WACCM and AURA MLS data, section 2 will now only present the ground-based microwave radiometers.

Old section 2 (Data sets) is renamed (Instruments and data sets) and splitted into two subsections: 2.1: Middle atmospheric water vapor radiometer and 2.2: Doppler wind radiometer. So the two used instruments are separately presented, which makes the structure more clear. Old subsections about SD-WACCM and Aura MLS are removed now.

(2)
If their measurement is only valid to 0.02 hPa (e.g. page 3), then they should cut off their plots at that level (e.g. Figures 1, 3, 6)

Yes, it is more convenient to cut off the $H_2O$ related plots at the upper measurement limit of 0.02 hPa, except for Figure 1, where we show the MIAWARA $H_2O$ time series together with the pressure level where the measurement response drops below 0.8.

As suggested from the referee, Figs. 3, 4, 6 and 8 are cut off at 0.02 hPa now.

(3)
End of abstract and beginning of Intro: They use "manifold" in adjacent sentences which seems awkward.

Thank you for the hint, we will replace the word manifold.

We changed the word "manifold" to "broad".

(4)
Page 2, line 2: "Latter analyzed"...?

We suggest the following new expression:

We changed "Latter analyzed..." to "..., who analyzed ..."

(5)
Page 2, line 23 either "a" or use plural

You are correct.

Now we use plural: ...ground-based water vapor oscillations...

(6)
Line 1 on page 3- what is this supposed to mean?

We use a much longer data set than the campaign-based study of Li et al. (2007). As supposed to this study, we are able to derive monthly mean wave amplitude characteristics in the sub-diurnal period range. That was the main point we wanted to express in this sentence.

Due the substantial manuscript changes in the introduction, this sentence was removed. In the discusion part of the results (Sect. 4.3) we take up the study of Li et al. (2007) in context with the possible (we are not sure) observations of low frequency inertia-gravity waves in our data sets.

(7)
Line 24-25 on page 4: again, a poorly expressed thought: I think I understand why the winter data are more usable- due to lower tropospheric humidity. But this sentence implies something else. Do they mean that the measurement response in winter is sufficiently high that they can use a time integration as short as 3 hours (as opposed to say, a day?). If so they need to express that more clearly.

Yes, this sentence was not expressed very clearly. Due to a lower amount of tropospheric water vapor in winter the signal from the middle atmosphere is less attenuated and we can use a shorter integration time for the observed $H_2O$ line spectrum at $22\,GHz$.

We now write: "During the winter months the tropospheric humidity is lower than during summer and in consequence the microwave signal from the middle atmosphere is

less attenuated by penetrating the troposphere to the ground-based receiver. Hence an integration of the signal of only 3 hours can be used to retrieve the $H_2O$ profiles. A conceptional parameter that is usually used to express the altitude dependent measurement sensitivity is the so-called measurement response."

(8)
Page 5,line 10: "To our knowledge and made efforts," ? "made efforts" is poor English.

Ok, thank you for the grammatical hint.

This sentence is removed in section 2. We explain and show results of our "efforts" that try to show that our observed oscillations are not related to artificial effects and thus a real atmospheric phenomenon now in the beginning of the discussion of the results in Sect. 4.3.

(9)
Page 12, line 25 "is capable of resolving"

You are correct.

Changed "...is capable to resolve" to "...is capable of resolving"

**3 Response to Referee #4**

**General remarks**

(1)
Whereas the authors wrote in the introduction at page 5 lines 10–12 To our knowledge and made efforts, artificial effects leading to the observed 18-hour variability can be excluded and therefore the wave is expected to be of atmospheric origin. We aim to report on findings based on middle atmospheric observations and model simulations. Revealing possible sources of an 18-hour inertia-gravity wave is beyond the scope of this paper. However, in the following they are only focusing on an 18-hour inertia gravity wave based on a single case study (Figs. 12 and 13). From the reviewer point, this generalization on all events is not valid because the difference between the observed period ($\sim$18 h) and the inertial period of 16.4 h as upper limit for the intrinsic period at the latitude of Bern (46.88°N) requires more or less at least constant background winds to get the Doppler shift of the intrinsic GW frequencies which has not been shown here.

The focus on the two wave events (with hodograph analysis) is removed by completely omitting the SD-WACCM data. The generalization from the 2 events was not valid. We

now only focus on our ground-based observations and let the explanation of the 18-hour wave open to some extent. In the discussion we now do not only focus on inertia-gravity waves but also on the possibility of a wave-wave interaction between the migrating diurnal tide and the quasi 2-day wave, which is a more likely explanation even in regard to the quite low vertical resolution of our microwave radiometers.

In the new Sect. 4.3 (Discussion) we add the argument about the background wind speed related to the inertia frequency: "...The main point would be to check if the vertical wavelengths are large enough for our microwave radiometer observations with a vertical resolution of more than 10 km to be able to see it. Further, in case of inertia-gravity waves with a ground related frequency of around 18 hours a specific background wind speed is required that reduces the actual intrinsic wave frequency (Doppler shifting) below the inertia frequency which is 16.44 h at the location of Bern."

(2)
An oscillation with a period of about 18 h can also be the result of a nonlinear wave-wave interaction of two waves, e.g. between quasi two day wave and the semidiurnal tide or between the semidiurnal and terdiurnal tide. This must be checked and considered as a possible reason for the obtained oscillations.

This is a very interesting hint. It could be that we observed a non-linear wave-wave interaction between the quasi 2-day wave and the migrating diurnal tide resulting in a westward traveling sum wave with periods around 16–18 hours, that behaves like a inertia-gravity wave (as stated in Lieberman et al. (2017)). So far we have not analyzed the quasi 2-day wave in our data for Bern, but this is planned in future. Within this paper revision it is virtually not possible for us to do this. In this paper we only focus on an interesting wave observation, but the clarification about the sources/causes has to be postponed.

We decided not only to discuss about our results in regard of inertia-gravity waves, but also in regard of such a non-linear wave-wave interaction in the new section 4.3 now. See also answer to major comment 1 by Referee #3.

(3)
In contrast to Figs 3 and 4, the SD-WACCM spectra show diurnal tidal waves with a poor spectral resolution, but no dominant oscillations between 15 and 21 h. It is surprising that there is such a similarity and significant correlation between the bandpass filtered wave amplitudes derived from MIAWARA and the corresponding water mixing ratio derived from SD-WACCM simulations (Figs 6–10). Can you comment this?

In Figure 10 only the correlations of individual amplitude profiles are shown. The plot somehow is misdirecting and does not show a temporal correlation between MIAWARA and SD-WACCM 18-hour amplitudes. We figured out, that the quite high correlation coefficients came from the good agreement between the model and observations at low

altitudes (below $0.1\,\mathrm{hPa}$), where no or only very small 18-hour wave amplitudes are present.

Figure 10 is removed and a comparison between MIAWARA and SD-WACCM in terms of the 18-hour variability is not meaningful. Anyway the whole SD-WACCM part of the original manuscript is canceled.

(4)
Please explain and/or improve the spectral resolution presented in Figs 3–5.

In Sect. 4.1 (Monthly mean $H_2O$ wave spectra) the spectral resolution is given as 1 hour. A even shorter spectral resolution does not make sense in our opinion because the sampling of the water vapor data is at 3 hour intervals. So the spectral resolution is already much shorter than the temporal resolution of the raw data.

Added the word "spectral" in Sect. 4.1 to make it more clear.

(5)
Please define the term "relative amplitudes" as used in Figs 6–10.

Thank you, it is important to make the term "relative amplitudes" more clear.

In the beginning of Sect. 4.2 (Temporal evolution of quasi 18-hour wave) we now write: "Absolute and relative wave amplitudes, which are calculated relative to the average water vapor mixing ratio at a pressure level over the investigated time period, are presented."

(6)
The case study (d) from 5–9 Dec 2015 (Fig 11) shows similar wave amplitudes between MIAWARA (water vapour), WIRA (u), and SD WACCM (u) and gives confidence that, with meridional winds from WIRA, a better wave estimation at the same location will be possible. In the frame of the used title focusing on 18 h waves, however, Fig. 4c show during this period only tides (12h, 24h) but nothing between 15–21 h

It is true that with meridional winds from WIRA a first estimation of the wave characteristic would be possible above the measurement site. The fact that our instrument see this wave leads also to the aspect that the 18-hour period waves must have a much larger vertical wavelength than the original stated range $\lambda_z < 6\,\mathrm{km}$. So to say, the SD-WACCM hodograph analysis resulted in impractical conclusions. The fact that Fig. 4c has no prominent amplitude peaks within the 15–21 h period range is due to the averaging over the entire month. The diurnal and semi-diurnal wave amplitudes were on average much larger in December 2015 than the 18-hour wave component. Unfortunately we can only use co-located WIRA data in December 2015 to compare with the water vapor observations.

Nothing changed accordant to the above comment 6.

(7)
The hodographs in Figs 13 and the derived possible characteristics of a monochromatic gravity wave are based on band pass filtered model simulations. It is not clear for the reviewer, how realistic are these simulated amplitudes, where the gravity waves are handled consistent a parametrization (see Page 6, lines 21–26). The cited papers of Baumgarten et al. (2015) and Li et al. (2007) used LIDAR data with a high resolution to estimate their hodographs. Please consider also a Stokes parameter analysis to get a more averaged GW description instead of the snapshot hodograph of a single monochromatic wave. Furthermore it is recommended to add the dispersion and Doppler equation to the wave parameter estimation to improve the readability.

We agree that the derived wave characteristics from filtered SD-WACCM simulations and their expressiveness was unclear in context to our observations. Probably other and more sophisticated model simulations (e.g. higher spatial resolutions) would be needed for a better comparison.

The complete hodograph analysis is not shown any more as stated before. Old Sect. 4.3.1 is removed and with it the numerical method part in Sect. 3 describing the hodograph method.

(8)
The AURA MLS temperatures and water vapour profiles are important for the MI-AWARA data as described in Sec 2.1 (page 4). However, at altitudes of about 0.1 hPa, where the observed 18 h oscillations have their maxima, the vertical resolution lies between delta h 5.5 and 6 km (see page 6, line 6) , so that only waves with vertical wavelengths larger than 2 x delta $h$ (11–12 km) can be resolved. From this point, the filtered temperature profiles with vertical wavelengths ¡ 6 km are questionable, at least above 0.1 hPa.

We agree that the filtered Aura MLS temperature profiles in regard of wavelengths below 6 km is problematic due to the too low vertical resolution.

We decided to completely remove the Aura MLS temperature analysis during the revision.

**Technical corrections**
(9)
Page 2 line 8 please add wind

Ok.

Page 2, line 8: We added "wind"

(10)
Page 7 line 13 bandpass

Ok.

Page 7, line 13: Changed "passband" to "bandpass".

(11)
Page 8 line 32 are given in the next Section.

OK.

Sentence is removed because it pointed to SD-WACCM results.

**References**

Li, T., She, C.-Y., Liu, H.-L., Leblanc, T., and McDermid, I. S. (2007). Sodium lidar observed strong inertia-gravity wave activities in the mesopause region over fort collins, colorado (41n, 105w). *J. Geophys. Res. Atmos.*, 112(D22). D22104.

Lieberman, R. S., Riggin, D. M., Nguyen, V., Palo, S. E., Siskind, D. E., Mitchell, N. J., Stober, G., Wilhelm, S., and Livesey, N. J. (2017). Global observations of 2 day wave coupling to the diurnal tide in a high-altitude forecast-assimilation system. *Journal of Geophysical Research: Atmospheres*, 122(8):4135–4149. 2016JD025144.

Nicolls, M. J., Varney, R. H., Vadas, S. L., Stamus, P. A., Heinselman, C. J., Cosgrove, R. B., and Kelley, M. C. (2010). Influence of an inertia-gravity wave on mesospheric dynamics: A case study with the poker flat incoherent scatter radar. *J. Geophys. Res. Atmos.*, 115(D3).

---

## Author Response (AR2)

**Quasi 18-hour wave activity in ground-based observed mesospheric water vapor over Bern, Switzerland**

*Martin Lainer, Klemens Hocke, Rolf Rüfenacht, and Niklaus Kämpfer*
* * *
**Response on ACPD paper acp-2016-1050 (Reconsider after major revision)**
* * *
**Color Code: Referee comments, Authors response, Link to relevant changes in manuscript**
* * *
We would like to thank all anonymous referees again and the editor for their comments to the first revised manuscript version. The new version (second revision) of our manuscript is revised accordant to the remaining minor and major issues announced by Reviewer #3 and Reviewer #4.

Please find our point by point response to those two Reviewers below. A marked-up manuscript version is provided in the end.

**1 Response to Referee #3**

**Major comment**

They have removed much of the questionable analyses and thus the paper is more carefully presented. It is less interesting as a result although I guess the 18 hour periodicity observation could merit a short note in ACP. I have one more very significant concern-namely their wind result, and specifically Figure 9b, is simply not credible. Consider that they have only 4 days of useable data with a 6 hour integration time. That means they have 16 points. And with 16 points, they are claiming to resolve about 5 oscillations of the 18 hour wave? This is very hard to believe even for a stationary amplitude. But to show time dependence like in Figure 9b? The uncertainties in the spectral domain have to be enormous. The slightest bit of non-stationarity, for example, will, I believe, completely invalidate their results. Further, a detailed look at Figure 9 shows it to not be consistent with Figure 4. Thus above 0.2 hpa, Figure 4 shows a white area meaning no data on Dec 5. So at the highest altitudes, they have less than 16 points. One suggestion is to limit themselves to the pressure domain from 0.5 to 1.0 hpa, where they have 7 days of data.

- We agree that the usable zonal wind data of just 4 days is rather short for the performed spectral analysis. In consequence we will remove the analysis related to Figure 9. Instead we now investigate a different dataset from WIRA, when it was deployed at the Observatoire de Haute-Provence in France in 2013. There

continuous zonal wind measurements for 17 days were performed covering the whole altitude range between 0.1–1 hPa without gaps. These wind measurements cannot be declared as co-located to MIAWARA and thus we omit a comparison of $H_2O$ and wind amplitudes. In total we have now 68 data points for the spectral analysis, which should be sufficient enough with a much smaller error compared to the previous results. A rough estimate of the pressure averaged (0.1–1 hPa) RMSE of the absolute zonal wind wave amplitudes revealed values between 3.3 and $6.6 \, \mathrm{m \, s^{-1}}$ over the investigated time period. The temporal mean RMSE of the zonal wind amplitudes is $5.4 \, \mathrm{m \, s^{-1}}$. After measuring the signal power of the WIRA zonal wind time series white Gaussian noise was added to reach a signal to noise ratio of 5 dB. The RMSE then was calculated from two spectral filter analyses of the noisy and original data set.

The new Figure 9 (WIRA absolute and relative 18h wave amplitudes) is now consistent with new Figure 4 (WIRA zonal wind data time series).

- A new zonal wind data set is introduced with 17 days of measurements in total. A new Figure 4 is included, which shows the data that is used to perform the spectral analysis shown in Figure 9a (absolute wave amplitudes) and b (relative wave amplitudes).

- Section 4.2 is adjusted and describes the results from the new WIRA data set and spectral analysis with the new error estimation. The text part belonging to the old analysis is removed.

- Abstract on page 1, line 8ff is adjusted: "18-hour oscillations in mid-latitude zonal wind observations from the microwave Doppler wind radiometer WIRA could be identified within the pressure range 0.1–1 hPa during an ARISE (Atmospheric dynamics Research InfraStructure in Europe) affiliated measurement campaign at the Observatoire de Haute-Provence (355 km from Bern) in France in 2013."

**Minor comments and technical/grammar correction**

(1)
On line 2, page 4, they state that they retrieved zonal and meridional winds, but show only the zonal component. What about the meridional component?

- The retrieval of meridional wind requires a much longer integration time than 6 hours (usually at least 24 hours) in the ground-based observations and is thus not usable for investigating an 18 hour oscillation. The 6 hours are enough to retrieve the zonal wind component. One reason is that the zonal wind in the middle atmosphere is much stronger than the meridional wind.

- Page 8, line 3: Changed "... meridional winds in sufficient high quality." to "... meridional winds in the temporal resolution required."

- Page 4 line 7: Here we added two sentences explaining why we do not show meridional winds in our study: "However the data quality of the meridional wind component retrieved from 6 hourly spectral line integrations is mediocre and thus not used in our study. In order to retrieve the meridional wind in the middle atmosphere a much longer integration time, like 24 h, is needed."

(2)
They mention meridional gradients in the H2O as possibly resulting from the edge of the polar vortex (line 10, page 8). I would argue that vertical gradients are equally if not more deserving of mention. They would not be expected to see oscillations in the H2O for pressures higher than about 0.2 hpa because the vertical gradient in the H2O becomes so weak. This means that the altitude variation of the 18 hour wave in H2O and in particular, a comparison with the wind results in Figure 9, is misleading. But then again, Figure 9b has bigger problems as per my top comment.

- We agree that it is worth to mention that the vertical gradient in water vapor is important for the observation of waves. Further we will remove the comparison between $H_2O$ and wind amplitudes. Figure 9 is a new Figure from a much larger zonal wind data set from the WIRA radiometer. This change is according to major comment 1.

- Page 8, line 14ff: Here we add the following: "Further vertical gradients in mesospheric water vapor are present and will influence the altitude region where $H_2O$ oscillations can be identified. It complicates the comparison between wind and $H_2O$ wave signatures."

- Section 4.2 (Temporal evolution of quasi 18-hour wave): The text parts that compared the MIAWARA and WIRA wave amplitudes in regard of Figure 9 are removed. We note here also:
  "A comparison of zonal wind and water vapor wave amplitudes could be misleading since the second highly depends on the vertical gradient. Thus such a comparison is not performed and the zonal wind analysis shown here has to be seen as an independent result and short part of the paper."

(3)
What are the vertical hash lines in Figures 2 and 3?

- The vertical and horizontal hash lines show the grid of the plot. So the vertical lines show the day of the month.

- Nothing changed.

(4)
Grammar: line 18, page 3: say "A typical range for the threshold for the ."

- Yes, your suggested expression sounds better. However, this paragraph was updated in order to clarify the averaging kernel and measurement response. The sentence you are talking about changed to the following:

- Page 3, line 18: "... To define a reliable altitude range for our retrieved data we use a typical range for the threshold for the measurement response between 60–80 %."

(5)
Grammar: line 22, page 3: "significantly"

- Yes, "significantly" is the right form here.

- Page 3, line 22: Changed "significant" to "significantly".

(6)
Page 6, line 26: delete "water vapor", saying "climatology as a priori information" is clear without redundancy

- Here we correct the wrong and misleading expression of "seasonally varying Aura MLS climatology". Actually we use a monthly mean zonal mean climatology from MLS between 2004 and 2010.

- The suggested removal of the expression "water vapor" is skipped, since the whole sentence changed.

- Page 6, line 26ff: "Further, we use a monthly mean zonal mean water vapor climatology based on Aura MLS v2.2 measurements between 2004 and 2010 as a priori information in the retrieval process, that are not expected to influence the observed $H_2O$ oscillations."

- $A_c$ is the a priori contribution which can be converted to the measurement response $M_r$ via $A_c = 100\% - M_r$.

- We introduce a new equation 1 in Section 4.3 to clarify the connection between a priori contribution $A_c$ and measurement response $(M_r)$.

**(8)**
Figure 11 in general- I am very confused. If the a priori is seasonally varying, and only seasonally varying, why do they get any kind of short term oscillations as shown in the Figure? Why not use monthly averaged a priori, then the question of short term (quasi 2 day) oscillations becomes moot.

- Sorry, the expression seasonally varying was misleading. Actually we use monthly mean zonal mean $H_2O$ climatology, that is based on Aura MLS v2.2 data between 2004 and 2010. The point why we get any kind of short term variability in Figure 11 is that we do not analyze the a priori data itself here, but the a priori contribution $(A_c)$. $A_c$ varies between every single $H_2O$ profile retrieval because it depends on the smoothing error of the measurements. This variability can also be seen in the white lines of Figure 1, which show the measurement response value of 80 %, which is a priori contribution $A_c$ of 20 %.

- In Section 4.3 (Discussion), page 6 line 26ff, we add a paragraph in order to give more detailed informations on the a priori contribution:
  "Although the a priori itself only varies monthly the measurement response as shown by the white horizontal lines in Fig. 1 varies on a much shorter time scale. This is due to the fact that the a priori contribution, which is the sensitivity of the retrieval to the a priori profile, depends on the actual smoothing error related to a single $H_2O$ profile retrieval with the integrated (in our case 3 hours) microwave line spectrum. To our knowledge it is not known how and if any short term variability of the measurement response effects the amount of retrieved water vapor. To clarify the terms a priori contribution $(A_c)$ and measurement response $(M_r)$ the conversion equation of these two quantities is given: ..."

- The caption of Figure 11 has been updated.

**(9)**
Continued of the above- line 31, page 6. This is unsubstantiated.

- This contextless sentence will be removed. More detailed information on the a priori contribution will be given in Section 4.3. This hopefully makes Figure 11 easier accessible.

- Page 6, line 31: This sentence is removed.

**2 Response to Referee #4**

**Minor revisions**

The revised manuscript mainly deals with observations of ground based mesospheric water vapour using the MIAWARA experiment near Bern (46.88° N) during 12 winter months from Oct 2014 – March 2015 and from Oct 2015 – March 2016. The authors found in 7 of these 12 months dominating oscillations with periods between 15 and 21 h. To interpret these oscillations, they used for case studies zonal winds from the Doppler wind radiometer WIRA at the same location. The combination of these datasets itself is valuable and suitable for publication in ACP. Compared to the previous version, this paper is nearly a new one focusing on own ground based measurements. I recommend to publish the manuscript after minor revision considering the following comments:

(1)
Page 3 Line 17: please explain "the area of the averaging kernel (the so-called measurements response)" in more detail or with a suitable reference. The measurement response is here given in percent in contrast to the WIRA (see Page 4, line 10)

- In case of the averaging kernels we now refer to the book of Rodgers (2000) and explain their meaning with regard to the measurement response. The term area of averaging kernel matrix is deleted.

- We now describe the measurement response in terms of percentage.

- Page 3, line 17ff: "In order to characterize a retrieved $H_2O$ profile the averaging kernel matrix can be used (Rodgers, 2000). This key quantity describes to what extent the retrieval is smoothing the true atmospheric state and how sensitive it is to the a priori profile (measurement response). To define a reliable altitude range for our retrieved data, we use a typical range for the threshold for the measurement response between 60–80 %."

- Page 4, line 10: Changed 0.8 to 80 %

(2)
Fig. 3 does not show the monthly time series for Winter 2015/2016 because it is identical with Fig. 2. However, in the present form these figures makes no sense and could be deleted, even if these both figures describes the data where the 18 hour oscillations appeared, because the authors didn't explain the observed variability and they didn't make any use of these figures in the discussion later on.

- Yes, this was a mistake. We will update the Figure 3 to show the monthly time series for winter 2015/2016. These figures were provided to show more raw data of our water vapor radiometer. This point has been claimed by one of the reviewers in the first revision stage. We think that the variability can be much better seen in the line plots than in the contour plot of Figure 1. For now we do not want to follow the advice to remove these figures.

- Page 13, Figure 3: Exchanged wrong Figure with the correct one, that shows the $H_2O$ time series for winter 2015/2016.

(3)
Page 3 Sect. 2.2 line 33: it is recommended to a add here a remark to lidar wind measurements with a higher vertical and temporal resolution as presented in the cited paper by Baumgarten et al., GRL, (2015) or in the paper by F.-J. Lbken, G. Baumgarten, J. Hildebrand and F. J. Schmidlin, Simultaneous and co-located wind measurements in the middle atmosphere by lidar and rocket-borne techniques, Atmos. Meas. Tech., 9, 3911-3919, doi:10.5194/amt-9-3911-2016, 2016, even if the application of the LIDAR wind technique requires a cloud free sky.

- We add here the remark of lidar wind measurements and refer to Baumgarten et al. (2015) and Lübken et al. (2016). We also note that the temporal and vertical resolution is higher than those from the WIRA instrument.

- Section 2.2, page 3, line 33ff: "Other techniques like rocket (Schmidlin, 1986) or lidar-based measurements (Lübken et al., 2016; Baumgarten et al., 2015) can provide wind data in this region with a higher vertical and temporal resolution than WIRA but suffer from high operational costs (rockets, lidar) or cloudy conditions (lidar). Meteorological rocket soundings are thus only suitable for short campaigns but not for continuous observations."

(4)
Page 5 line 5 : 0.25 – 0.35

- You are correct, this was a typo.

- Page 5, line 5: Corrected to "0.25 - 0.35 ppm"

(5)
Page 5 Sect 4.2: line 34 please add here a reference to Fig. 9a, because this is an important result.

- Figure 9 completely changed. We now show a different WIRA data analysis from 25th of January to 10th of February 2013. Further we now do not compare the zonal wind amplitudes to MIAWARA $H_2O$ wave amplitudes. This is done with regard to the other reviewers first major comment. A reference to Fig 9a is thus redundant.

- Nothing changed.

(6)
The new discussion in Sect 4.3 contains two parts which could be separated in two subsections.

- It is not clear to us which two parts the reviewer is talking about. By now we have not changed anything. Also the discussion part is separated in various paragraphs which imply a kind of separation already.

- Nothing changed.

(7)
Page 7 line 34 below the period of 16.44 h corresponding to the inertia frequency at the latitude of Bern

- Thanks for this suggestion. It sounds better and we changed the expression accordingly.

- Page 7, line 34: "...below the period of 16.44h corresponding to the inertia frequency at the latitude of Bern."

- We deleted "maybe" in this sentence and changed the word order of "...zonal wind and water vapor..."

[revised manuscript text omitted]

---

## Author Response (AR3)

**Quasi 18-hour wave activity in ground-based observed mesospheric water vapor over Bern, Switzerland**

*Martin Lainer, Klemens Hocke, Rolf Rüfenacht, and Niklaus Kämpfer*
* * *
**Response on ACPD paper acp-2016-1050 (Publish subject to minor revisions)**
* * *
**Color Code:** Referee comments, Authors response, Link to relevant changes in manuscript
* * *
We would like to thank referee #3 and the editor for their comments to the latest revised manuscript version. The new version of our manuscript is revised accordant to the remaining minor and editorial issues.

We note also that we corrected a color bar related error we detected in Figures 5, 6 and 11. This resulted in slight changes to the wave amplitudes and improved the visualization of the contour plots (more contour levels). The message of the paper did not change according to this correction. We adjusted the following:

- Page 5, Section 4.1, Line 15: Instead of 0.25–0.35 ppm the amplitudes are now in the range 0.2–0.3 ppm.

- Page 7, Section 4.3, Line 14-15: It is now correct to state that "All three months have mean wave amplitudes below 7 % at pressure level above..."according to the updated (corrected) Figure 11.

**1 Response to Referee #3**

**Minor comment 1**

First, the paragraph beginning on page 8, line 31 was a bit confusing and scattered. It starts out with a mention of the polar vortex and then includes comments about vertical gradients in H2O, a topic which does not seem to have much to do with the polar vortex. What are they trying to say with this paragraph? Is it that a direct comparison with wind data is complicated? OK, if so, then say so more directly and then follow the general statement with specific reasons why.

- The part of this paragraph (page 8, line 31ff) dealing with vertical gradients is indeed misleading and confusing. This part is canceled now and we tried to more clearly emphasize the complication you mentioned.

- Page 8, Line 31ff reads now: "Another complication in the comparison between wind and $H_2O$ wave signatures comes from oscillations in $H_2O$ that may be caused by the polar vortex edge moving across the observation site. Across the polar vortex edge large meridional gradients in $H_2O$ tracer concentrations exist. A regular movement of the whole vortex could therefore trigger oscillations in atmospheric $H_2O$ profile measurements. Indeed we find such oscillations of the polar vortex edge during winter above Bern, but the dominant period is 24 h in the mesosphere (0.01–1 hPa). We could not find any connection to an 18-hour period we are focusing on in this study."

**Minor comment 2**

They could also helpfully add, someplace in the text- perhaps here or in the Conclusion, a simple comment that at very least the wind observations do provide an additional constraint for 3D models which might be trying to reproduce their data. If this wave has not yet been simulated or discussed in models such as WACCM or ECHAM, that would be useful information to know.

- Yes, we think this could be a nice hint in our conclusion part. We mention now WACCM and ECHAM with regard to the constraint to wind observations. A new reference (Huang et al. 2013) has been included. They observed similar wave periods ($\sim 16$ h) with radar observations. Finally we propose a useful exercise to validate the representation of non-linear wave-wave couplings in those models. In the discussions we already mentioned the NOGAPS ALPHA model system and the related study by Lieberman et al. (2017).

- The last paragraph of our conclusions reads now in the updated version: "It has been shown that the Doppler wind radiometer WIRA is capable to resolve sub-diurnal oscillations in the zonal wind component. The quality of the meridional wind measurements have a potential for improvement and could contribute to wave characteristic analyses in the near future. Quasi 18-hour oscillations were detected in the WIRA zonal wind data set for a period of about 17 days. There are also meteor radar based measurements (Huang et al., 2013), where similar wave periods ($\sim 16$ h) are observed. At the very least the different wind observations could provide an additional constraint for 3-dimensional model simulation studies achievable with for instance WACCM (Whole Atmosphere Community Climate Model) or ECHAM. A useful exercise could aim at validating the correct representation of non-linear wave-wave couplings in different models involving tides and planetary waves like the quasi 2-day wave."

**Editorial comments**

(1)
Page 6, line 11. They should take care to emphasize that this is a qualitative observation only. Then, with also correcting the English: "Qualitatively, this looks like the same behavior as revealed by the water vapor analysis"

- We surely agree that this is a qualitative observation and correct accordingly.

- Page 6, Line 11: Changed to: "Qualitatively, this looks like the same behavior as revealed by the water vapor analysis."

(2)
Page 9, line 28: straightened out is too informal for the academic literature. Suggest: clarified

- Page 9, Line 28: Replaced "straightened out" with "clarified" as suggested by the reviewer comment.

[revised manuscript text omitted]